EMBO
Molecular Medicine

# CUEDC2 modulates cardiomyocyte oxidative capacity by regulating GPX1 stability

Zhao Jian[1], Bing Liang[2], Xin Pan[2], Guang Xu[2], Sai-Sai Guo[2], Ting Li[2], Tao Zhou[2], Ying-Bin Xiao[1,*] & Ai-Ling Li[2,**]

## Abstract

The irreversible loss of cardiomyocytes due to oxidative stress is the main cause of heart dysfunction following ischemia/reperfusion (I/R) injury and ageing-induced cardiomyopathy. Here, we report that CUEDC2, a CUE domain-containing protein, plays a critical role in oxidative stress-induced cardiac injury. $Cuedc2^{-/-}$ cardiomyocytes exhibited a greater resistance to oxidative stress-induced cell death. Loss of CUEDC2 enhanced the antioxidant capacity of cardiomyocytes, promoted reactive oxygen species (ROS) scavenging, and subsequently inhibited the redox-dependent activation of signaling pathways. Notably, CUEDC2 promoted E3 ubiquitin ligases tripartite motif-containing 33 (TRIM33)-mediated the antioxidant enzyme, glutathione peroxidase 1 (GPX1) ubiquitination, and proteasome-dependent degradation. Ablation of CUEDC2 upregulated the protein level of GPX1 in the heart significantly. Strikingly, *in vivo*, the infarct size of $Cuedc2^{-/-}$ heart was significantly decreased after I/R injury, and aged $Cuedc2^{-/-}$ mice preserved better heart function as the overall ROS levels in their hearts were significantly lower. Our results demonstrated a novel role of CUEDC2 in cardiomyocyte death regulation. Manipulating CUEDC2 level might be an attractive therapeutic strategy for promoting cardiomyocyte survival following oxidative stress-induced cardiac injury.

**Keywords** ageing-induced cardiomyopathy; antioxidant capacity; CUEDC2; GPX1; ischemia/reperfusion injury
**Subject Category** Cardiovascular System

## Introduction

In the setting of acute myocardial infarction (MI), a leading cause of death worldwide (Nabel & Braunwald, 2012), oxidative stress is purported to play a critical role in cardiomyocyte loss developed in the period of ischemia and following reperfusion (Giordano, 2005).

In addition to this acute injury induced by oxidative stress, high levels of ROS accelerate cardiomyocyte death and play critical roles in myocardial remodeling, heart failure, and ageing-induced cardiomyopathy (Giordano, 2005). Therefore, strengthening reactive oxygen species (ROS) scavenging is an attractive strategy to alleviate myocardial injury and enhance heart function (Eltzschig & Eckle, 2011).

Among the well-defined antioxidant pathways, glutathione peroxidases (GPXs) play a pivotal role in the cellular defense against oxidative stress by reducing $H_2O_2$ and a wide range of organic hydroperoxides (Lubos *et al*, 2011). In the heart, GPX1 is one of the most important antioxidant enzymes for the detoxification of $H_2O_2$. The ablation of GPX1 results in robust oxidative stress and more severe heart dysfunction following I/R injury (Forgione *et al*, 2002), highlighting the therapeutic potential of GPX1 regulation in ischemic heart. In fact, transgenic mice overexpression of GPX1 exhibited higher resistance to I/R injury (Yoshida *et al*, 1996). However, as a powerful antioxidant enzyme, GPX1 overexpression over-quenches intracellular ROS, which is required for some pro-survival kinases such as extracellular-regulated kinase (ERK) 1/2 activation (Autheman *et al*, 2012). This might be negative for some physiological function and discount the beneficial effects of hypoxia preconditioning. In addition, GPX1 is ubiquitously expressed in all tissues and the overexpression of GPX1 could cause insulin resistance and obesity (McClung *et al*, 2004). Therefore, direct overexpressing GPX1 might not be an ideal therapeutic strategy for heart diseases. Modulating GPX1 level indirectly might win more effective protection on I/R injury, without interfering with the physiological function. Unfortunately, although some factors can affect GPX1 level by regulating GPX1 mRNA stability (Jornot & Junod, 1997; Zhang *et al*, 2005), the regulation of GPX1 expression and activity remains poorly defined.

We have previously reported that CUE domain containing 2 (CUEDC2) not only promotes the ubiquitination and degradation of the progesterone receptor (Zhang *et al*, 2007; Pan *et al*, 2011) but also plays indispensable roles in inflammation and cancer (Li *et al*, 2008; Gao *et al*, 2011). However, the function of CUEDC2 remains largely undefined. Here, we demonstrated that CUEDC2 is important for E3 ligases tripartite motif-containing 33 (TRIM33)-mediated

1 Institute of Cardiovascular Surgery, Xinqiao Hospital, Third Military Medical University, Chongqing, China
2 State Key Laboratory of Proteomics, Institute of Basic Medical Sciences, National Center of Biomedical Analysis, Beijing, China
*Corresponding author. Tel: +86 23 68755607; Fax: +86 23 68755607; E-mail: xiaoyb@tmmu.edu.cn
**Corresponding author. Tel: +86 10 66930344; Fax: +86 10 68186281; E-mail: alli@ncba.ac.cn

GPX1 proteasome-dependent degradation. Loss of CUEDC2 promoted reactive oxygen species (ROS) scavenging and enhanced the antioxidant capacity of cardiomyocytes, leading to reduced oxidative stress and improved cardiomyocyte survival during either acute or chronic oxidative stress (i.e., I/R injury and ageing).

## Results

### CUEDC2 in the heart is degraded upon ischemic stimulation

In order to examine the biological roles of CUEDC2 *in vivo*, we firstly tested the tissue distribution of CUEDC2 protein in mice. We found that CUEDC2 was expressed in various organs including the brain, heart, thymus, and spleen (Fig 1A). Due to the particularly high protein level of CUEDC2, we were attracted to investigate the potential role of CUEDC2 in the heart. Therefore, we tested the protein level of CUEDC2 in the heart upon acute I/R injury or chronic overload stress, which were the most common pathophysiological stress in cardiovascular diseases. CUEDC2 was not changed under chronic stress even 2 weeks after aortic banding (Appendix Fig S1). Interestedly, after the onset of reperfusion following 30-min ischemia, the protein levels of CUEDC2 were gradually decreased (Fig 1B), but this did not occur under 30-min ischemia only (Appendix Fig S2A). This strongly suggested that CUEDC2 might play an important role under acute I/R injury. We confirmed that this decrease was specific, as the protein levels of another CUE domain-containing protein family member, CUEDC1, were not altered during reperfusion (Fig 1B). This decrease in CUEDC2 protein level was confirmed in neonatal mouse cardiomyocytes treated with hypoxia/reoxygenation (H/R), a condition that mimicked I/R *in vitro* (Zhang *et al*, 2016) (Fig 1C and Appendix Fig S2B). After we examined the mRNA level during I/R injury, we found that *Cuedc2* mRNA level was not changed (Appendix Fig S2C), indicating that the downregulation of CUEDC2 might be due to post-translational modification. And we further found that treatment with the proteasome inhibitor, MG-132, could essentially abolish this decrease (Appendix Fig S2D), suggesting that the decrease in CUEDC2 protein level was seemly mediated through proteasome-dependent degradation.

To test whether CUEDC2 degradation is also present in myocardial ischemia patients, we examined the CUEDC2 protein levels in samples from patients suffered from acute myocardial infarction (the patients' clinical information is listed in Appendix Table S1). Compared with that observed in the non-ischemic distant zone (DZ) and normal controls, where cardiomyocytes do not suffer from ischemia, the protein level of CUEDC2 was significantly reduced in the ischemic border zone (BZ), where cardiomyocytes struggle to survive under ischemia (Fig 1D and Appendix Fig S3). This supports the notion that ischemic stimulation could induce the degradation of CUEDC2 in the human heart.

### Ablation of CUEDC2 decreases ROS level and inhibits redox-dependent pathways under I/R injury

In an effort to investigate the roles of CUEDC2 degradation in response to ischemic stimulation, we generated *Cuedc2*$^{-/-}$ mice by homologous recombination shown in Appendix Fig S4A and as

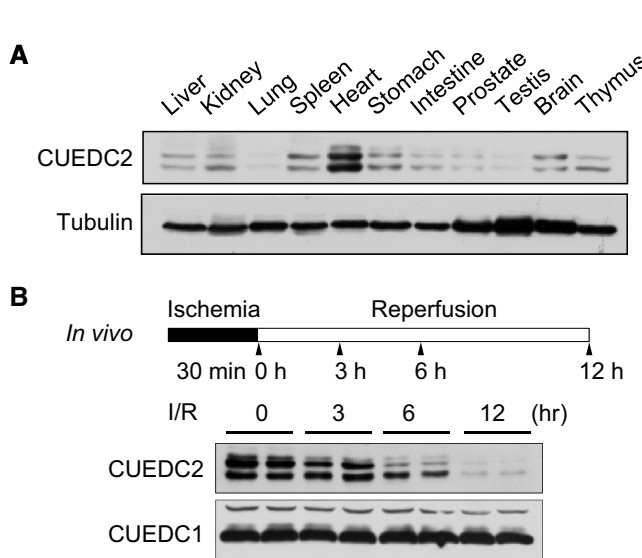

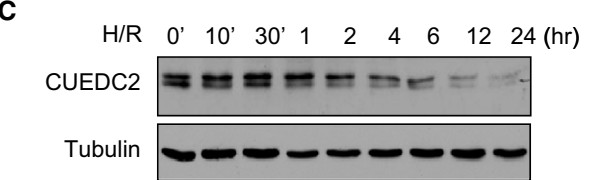

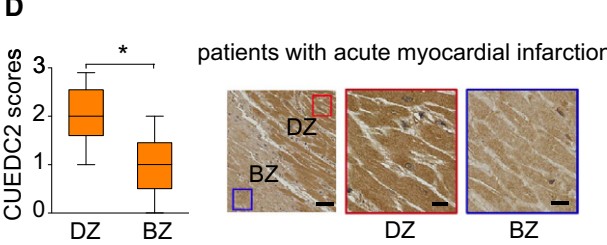

**Figure 1. CUEDC2 degradation was induced by ischemia/reperfusion in cardiac tissue.**

A   CUEDC2 protein levels in mouse tissues. Adult (12-week-old) mouse was sacrificed and the total protein was extracted from the liver, kidney, lung, spleen, heart, stomach, intestine, prostate, testis, brain, and thymus.

B   (top) *In vivo* cardiac ischemia/reperfusion model protocol (30-minute ischemia followed by indicated times of reperfusion). (bottom) Mice were subjected to reversible ischemia *in vivo* for 30 min and reperfused for different time periods. Protein was extracted from the area at risk of heart at the indicated time points and subjected to immunoblot analysis of CUEDC2 and CUEDC1. Representative results were shown from 2 mice at each time point, and experiments were repeated for three times.

C   Isolated primary rat neonatal cardiomyocytes were subjected to hypoxia with serum-free medium for 6 h and reoxygenated accompanying adding serum back for different time period *in vitro* and the protein level of CUEDC2 was tested at different time points during reoxygenation. Each experiment was repeated for three times.

D   CUEDC2 protein level was plotted using the immunohistochemical scores as described in methods. (left) Plot of CUEDC2 scores in each BZ and DZ (*n* = 9 patients). (right) Representative images from immunohistochemical staining for CUEDC2 in the tissues from patients who suffered acute myocardial infarctions. The boxed areas in the left images are magnified in the middle and right images. DZ, distant zone; BZ, border zone. Scale bars, 250 μm (left), 50 μm (middle and right). *$P$ = 0.001. Data were shown as mean ± SEM and analyzed by paired *t*-test.

Source data are available online for this figure.

reported recently (Chen *et al*, 2014). The gene knockout was confirmed by northern blot, and CUEDC2 protein expression was not detected in the heart of homozygous $Cuedc2^{-/-}$ mice (Appendix Fig S4B and C). The ratios of heart weight to body weight (HW/BW), showing the heart grows in relation to the whole body, were indistinguishable between $Cuedc2^{-/-}$ mice and that of their wild-type (WT) littermates (Appendix Fig S4D). Hearts of $Cuedc2^{-/-}$ mice appeared normal based on histological analysis (Appendix Fig S4E). Additionally, the expression of collagen and fetal genes, which are indicators of defects in the adult heart (Aurora *et al*, 2012), were similar between $Cuedc2^{-/-}$ and WT mice (Appendix Fig S4F). Moreover, transthoracic echocardiography showed no significant differences in cardiac chamber sizes and

heart function between adult $Cuedc2^{-/-}$ and WT control mice (Appendix Table S2). These data demonstrate that the ablation of CUEDC2 has no apparent impact on cardiac development and physiological function.

To examine the possible role of CUEDC2 degradation under ischemic stimulation, we treated neonatal cardiomyocytes derived from $Cuedc2^{-/-}$ mice and WT mice with $H_2O_2$ in culture. Compared with the WT controls, a significantly higher percentage of $Cuedc2^{-/-}$ cardiomyocytes survived the treatment (Fig 2A and B).

I/R-induced injury, such as cardiomyocyte death, is primarily triggered by the generation of massive amounts of ROS (Giordano, 2005). As the ablation of CUEDC2 mitigated $H_2O_2$-induced cardiomyocyte death, we reasoned that the intracellular ROS levels

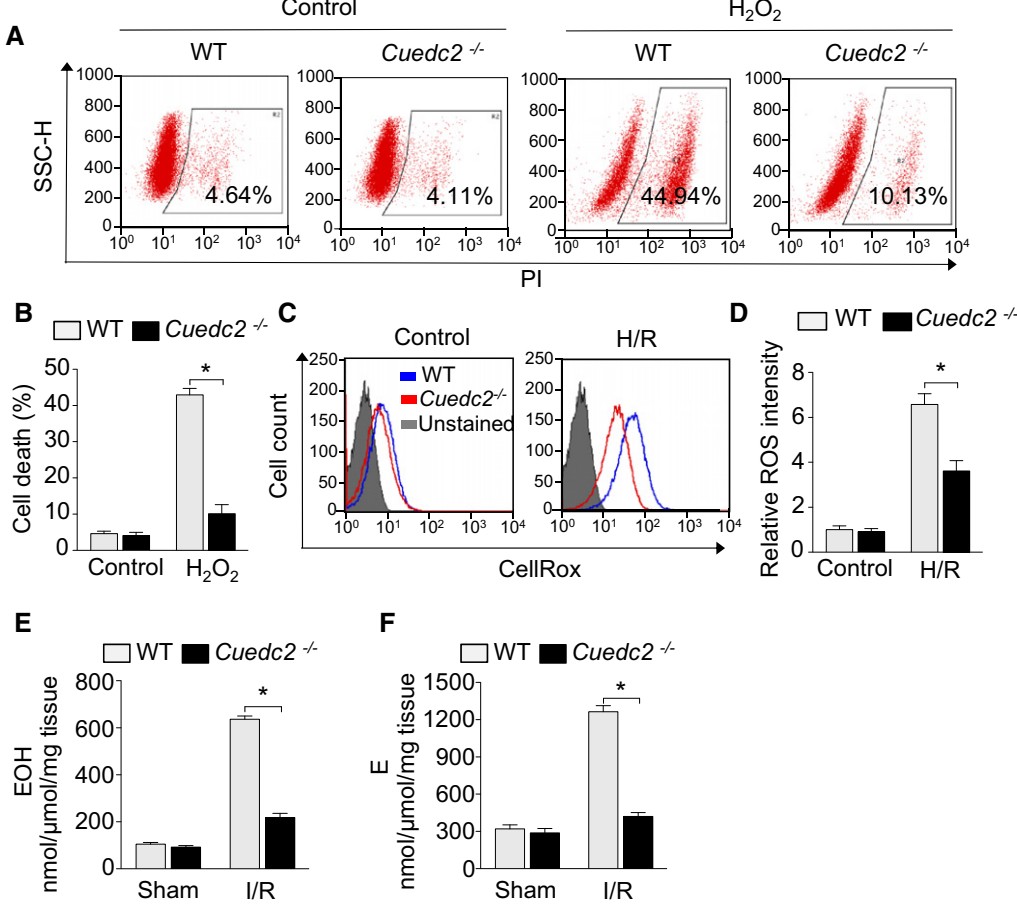

**Figure 2. Ablation of CUEDC2 promotes cardiomyocyte survival by enhancing antioxidant capacity.**

A  Primary mouse cardiomyocytes were treated with $H_2O_2$ (1 mM) for 3 h and cardiomyocyte viability was determined by flow cytometry. Representative results of flow cytometry analysis.

B  Quantitative analysis of PI-positive cells treated with $H_2O_2$ was shown (42.9 ± 3.1% in wild-type group vs 10.1 ± 4.4% in $Cuedc2^{-/-}$ group, *P = 0.0004, n = 5 wells per group and repeated for three times).

C  Neonatal mouse cardiomyocytes were subjected to hypoxia for 6 h and then reoxygenation for 30 minutes. Cardiomyocytes were then stained with 5 μM CellROX[®] Deep Red Reagent and analyzed by flow cytometry. Representative results of flow cytometry analysis are shown, and all experiments were repeated for three times.

D  ROS levels were quantitated and summarized. *P = 0.0108, n = 3 wells per group and repeated for three times.

E  HPLC detection of a superoxide probe oxidized dihydroethidium (DHE) product in sham and I/R injury heart tissue (30-min ischemia followed by 30-min reperfusion), 2-hydroxyethidium (EOH), a specific product for superoxide anion radical. *P < 0.0001, n = 5 in each group.

F  In the same samples of (E), HPLC detection of ethidium (E), oxidized by other reactive oxygen species such as $H_2O_2$ (mainly) and ONOO. *P = 0.0001, n = 5 in each group.

Data information: Data were shown as mean ± SEM and analyzed by unpaired *t*-test.

might be lower in $Cuedc2^{-/-}$ cardiomyocytes during reperfusion. As expected, the induction of ROS was significantly reduced in $Cuedc2^{-/-}$ cardiomyocytes by hypoxia/reoxygenation (H/R) (to mimic I/R in vitro) compared with that of WT cardiomyocytes (Fig 2C and D). Furthermore, by treating cardiomyocytes with menadione, which inhibits mitochondrial respiratory chain activity and induces high levels of ROS (Loor et al, 2010), we found that the levels of total ROS were much lower in $Cuedc2^{-/-}$ cardiomyocytes (Appendix Fig S5A and B). We further examined the cell death of cardiomyocytes under H/R treatment and found that cell death significantly increased in both WT and $Cuedc2^{-/-}$ cardiomyocytes. But the death of $Cuedc2^{-/-}$ cardiomyocytes was significantly lower than that of WT cardiomyocytes (Appendix Fig S5C).

To further quantitatively assess the production of ROS and oxidative stress in the heart under I/R injury, ROS were measured by the detection of DHE oxidation product EOH (2-hydroxyethidium), which is a marker of superoxide generation, and ethidium (E), which is a marker of $H_2O_2$ primarily and other ROS, with high-performance liquid chromatography (HPLC) (Puente et al, 2014). We found that the generation of superoxide, $H_2O_2$, and other ROS

were decreased in $Cuedc2^{-/-}$ heart, and $H_2O_2$ and other ROS were decreased more significantly (Fig 2E and F). These data indicate that the ablation of CUEDC2 enhanced ROS scavenging in cardiomyocytes.

Oxidative stress in vivo activates various signaling pathways, such as the MAPK pathway (Burgoyne et al, 2012), to induce myocardial injury. It is well established that the I/R-induced activation of c-Jun N-terminal kinase (JNK) is a modulator of cardiomyocyte survival decisions during oxidative stress (Liu et al, 2009; Wei et al, 2011). To examine the potential impact of CUEDC2 depletion on JNK activation, $Cuedc2^{-/-}$ and WT mice were subjected to I/R. JNK phosphorylation was significantly reduced in the hearts of $Cuedc2^{-/-}$ mice compared with those of WT mice (Fig 3A). Similarly, the activation of p38 MAPK, another kinase involved in I/R injury, was also dramatically decreased in $Cuedc2^{-/-}$ hearts (Fig 3A). However, the activation of ERK1/2 was not affected by CUED2 ablation (Appendix Fig S6). Consistent with these findings, the activation of JNK was essentially abolished in cultured $Cuedc2^{-/-}$ cardiomyocytes following H/R stimulation (Fig 3B).

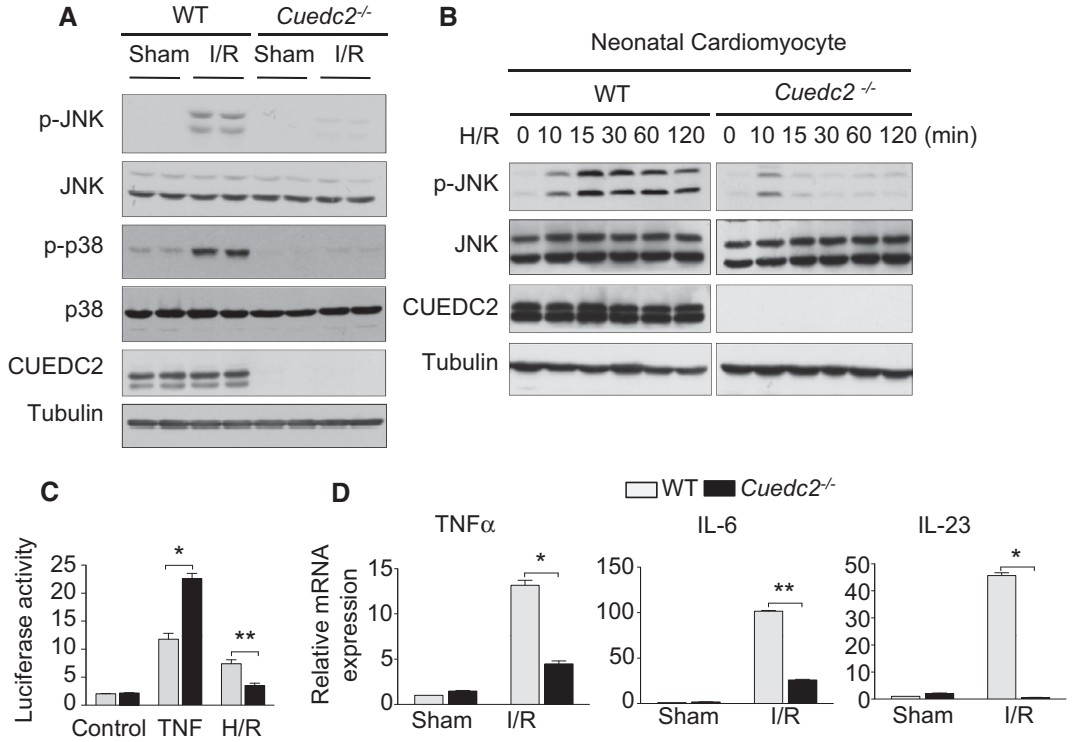

**Figure 3.  Loss of CUEDC2 inhibits ROS and redox-dependent pathways induced by oxidative stress.**

A   Protein was extracted from the area at risk of left ventricle from wild-type or $Cuedc2^{-/-}$ mice with 30-minute ischemia followed by 30-minute reperfusion and subjected to immunoblot analysis. Representative results from 2 mice are shown, and each experiment was repeated for three times.

B   Neonatal mouse cardiomyocytes were subjected to hypoxia for 6 h, followed by the indicated times of reoxygenation, before the protein was extracted and subjected to immunoblot analysis.

C   Luciferase assay of WT or $Cuedc2^{-/-}$ MEFs transiently transfected with an NF-κB-responsive luciferase reporter and then, 1 d later, treated for 6 h with TNF (10 ng ml$^{-1}$; TNF) or for H/R. * $P$ = 0.0032, $n$ = 3 per group, **$P$ = 0.013, $n$ = 3 per group.

D   Wild-type and $Cuedc2^{-/-}$ mice were treated with a sham or I/R operation in vivo. Three hours after the onset of reperfusion, the mRNA levels of tumor necrosis factor α (TNF-α), interleukin 6 (IL-6), and interleukin 23 (IL-23) were tested. *$P$ = 0.0005, **$P$ = 0.0001, $n$ = 3 per group.

Data information: Data were shown as mean ± SEM and analyzed by unpaired $t$-test.
Source data are available online for this figure.

As we previously reported, CUEDC2 could inhibit nuclear factor-kappa B (NF-κB) activation by decreasing phosphorylation and activation of IκB kinase (IKK) (Li *et al*, 2008). While NF-κB plays an important role in I/R injury (Nakano *et al*, 2005), we next detected the activity of NF-κB in the hearts of *Cuedc2*$^{-/-}$ and WT mice. Consistent with previous reports, compared with sham-operated mice, the binding activity of NF-κB to target gene was upregulated in the ischemic myocardium of I/R-treated mice (Appendix Fig S7A and B). Also, the transcriptional activity of NF-κB was increased under I/R injury (Fig 3C). Accordingly, mRNA levels of NF-κB target genes, such as tumor necrosis factor α (TNF-α), interleukin 6 (IL-6), and interleukin 23 (IL-23), were all upregulated under I/R stimulation (Fig 3D) (primers for Q-PCR was shown in Appendix Table S3). However, binding and transcriptional activities and the upregulation of these inflammatory factors were all inhibited in the CUEDC2-deficient heart following I/R, indicating that activation of NF-κB was inhibited when CUEDC2 deleted in this I/R context (Fig 3C and D). Taken together, the ablation of CUEDC2 promotes cardiomyocyte survival by decreasing the I/R-induced ROS generation and subsequently inhibiting the redox-mediated signaling pathways associated with I/R injury.

### *Cuedc2* deletion enhances the antioxidant potential of cardiomyocytes by upregulating GPX1

Superoxide dismutases (SODs) and glutathione peroxidases (GPXs), which catalyze the reaction of $O_2^{\cdot-}$ to $H_2O_2$ and $H_2O_2$ to $H_2O$, respectively, are the main enzymes involved in ROS detoxification. Therefore, we examined the protein levels of these enzymes in the hearts of *Cuedc2*$^{-/-}$ and WT mice. Our data indicated that the protein levels of GPX1 were significantly higher in *Cuedc2*$^{-/-}$ heart than in WT hearts, whereas manganese superoxide dismutase (MnSOD) and glutathione peroxidase 5 (GPX5), another paralog of GPX1, were expressed at similar levels in *Cuedc2*$^{-/-}$ and WT hearts (Fig 4A). In addition, the expression levels of GPX1 were inversely correlated with the expression levels of CUEDC2 following I/R injury, suggesting that CUEDC2 negatively modulated the expression level of GPX1 (Fig 4B). Moreover, we examined the GPX1 in the same tissue microarray used in Fig 1D, and we found that the protein level of GPX1 was significantly increased in the ischemic border zone (BZ) (Fig 4C). Importantly, when the expression of GPX1 was knocked down, knocking out CUEDC2 failed to protect against $H_2O_2$-induced cardiomyocyte death (Fig 4D and E). Similarly, in the absence of GPX1, the loss of CUEDC2 could no longer inhibit $H_2O_2$-induced ROS production (Fig 4F). These results indicate that GPX1 upregulation is the downstream molecular effector underlying the protective effects of CUEDC2 deletion.

### CUEDC2 destabilizes GPX1 by facilitating its ubiquitin-proteasome-dependent degradation

As the decrease in GPX1 was not at the transcriptional level (Appendix Fig S8A) and previously reported that CUEDC2 promotes the ubiquitination and degradation of the progesterone receptor (PR) and estrogen receptor-α (ER-α) in breast cancer cells. To explore the possible effects of CUEDC2 on GPX1, we first examined the stability of GPX1 protein in WT and *Cuedc2*$^{-/-}$ mouse embryonic fibroblasts (MEFs). Indeed, after protein synthesis was inhibited by cycloheximide (CHX), GPX1 protein was much more stable in *Cuedc2*$^{-/-}$ MEFs than in WT controls. The half-life of GPX1 protein was approximately 3 h in WT MEFs, whereas this extended to approximately 15 h when CUEDC2 was ablated (Fig 5A). Furthermore, rescuing the expression of CUEDC2 in *Cuedc2*$^{-/-}$ MEFs led to a decrease in GPX1 amounts. Importantly, the effect of CUEDC2 on GPX1 was inhibited in the presence of the proteasome inhibitor MG-132 (Fig 5B), suggesting that the ubiquitin-proteasome pathway may be required for the CUEDC2-mediated downregulation of GPX1 protein. In addition, MG-132 could stabilize GPX1 protein in WT neonatal cardiomyocytes, but failed to do so in *Cuedc2*$^{-/-}$ cardiomyocytes, suggesting that CUEDC2 represents a major pathway for stimulating GPX1 ubiquitination-dependent degradation in cardiomyocytes (Appendix Fig S8B). Indeed, when we co-transfected vectors expressing GPX1, CUEDC2, and ubiquitin into HEK293T cells, GPX1 was ubiquitinated, and the overexpression of CUEDC2 markedly increased the ubiquitination of GPX1 (Fig 5C). Consistent with these findings, the ablation of CUEDC2 resulted in a decrease in GPX1 ubiquitination levels *in vivo* (Fig 5D). Moreover, we found that CUEDC2 could interact with GPX1 and CUE domain was indispensable for the interaction between CUEDC2 and GPX1 (Fig 5E) and that their interaction was more robust in the presence of MG-132 (Appendix Fig S9A). Moreover, the overexpression of the CUE domain deleted CUEDC2 mutant could not lead to the decrease in GPX1 level (Appendix Fig S9B). These results indicate that the interaction between CUEDC2 and GPX1 might be the underlying molecular basis for CUEDC2-mediated GPX1 suppression.

To further identify the possible E3 ubiquitin ligase of GPX1, we transiently overexpressed GPX1 in HEK293T cells, immunoprecipitated the GPX1 protein complex, and analyzed by mass spectrometry (Appendix Fig S10 and Appendix Table S4). We successfully identified two potential E3 ubiquitin ligases interacting with GPX1, tripartite motif-containing 33 (TRIM33) and F-box and WD repeat domain containing 7 (FBXW7). After we overexpressed TRIM33 in different level, we found that the protein level of GPX1 decreased accordingly (Fig 6A), while FBXW7 had no effect on GPX1 protein level (Fig 6B). Importantly, after we mutated the ubiquitination-associated RING motif of TRIM33, it could not promote GPX1 ubiquitination, indicating that TRIM33 is an E3 ubiquitin ligase for GPX1 (Fig 6C). Interestedly, the protein level of GPX1 could not be downregulated by TRIM33 when we silenced the expression of CUEDC2 (Fig 6D). After we knocked down the expression of TRIM33 in primary cardiomyocytes and the inverse relationship between CUEDC2 and GPX1 was abrogated (Appendix Fig S11A), and CUEDC2 did not affect the interaction of GPX1 with its E3 ligase TRIM33 (Appendix Fig S11B). Taken together, these results indicate that CUEDC2 is critical for TRIM33-mediated GPX1 ubiquitin-dependent degradation.

### *Cuedc2*-deficient mice are more resistant to ischemic injury *in vivo*

To study the antioxidant effect of CUEDC2 ablation *in vivo*, we next subjected *Cuedc2*$^{-/-}$ mice and their WT littermates to I/R injury. The ratios of the area at risk to left ventricle area (AAR/LV) did not

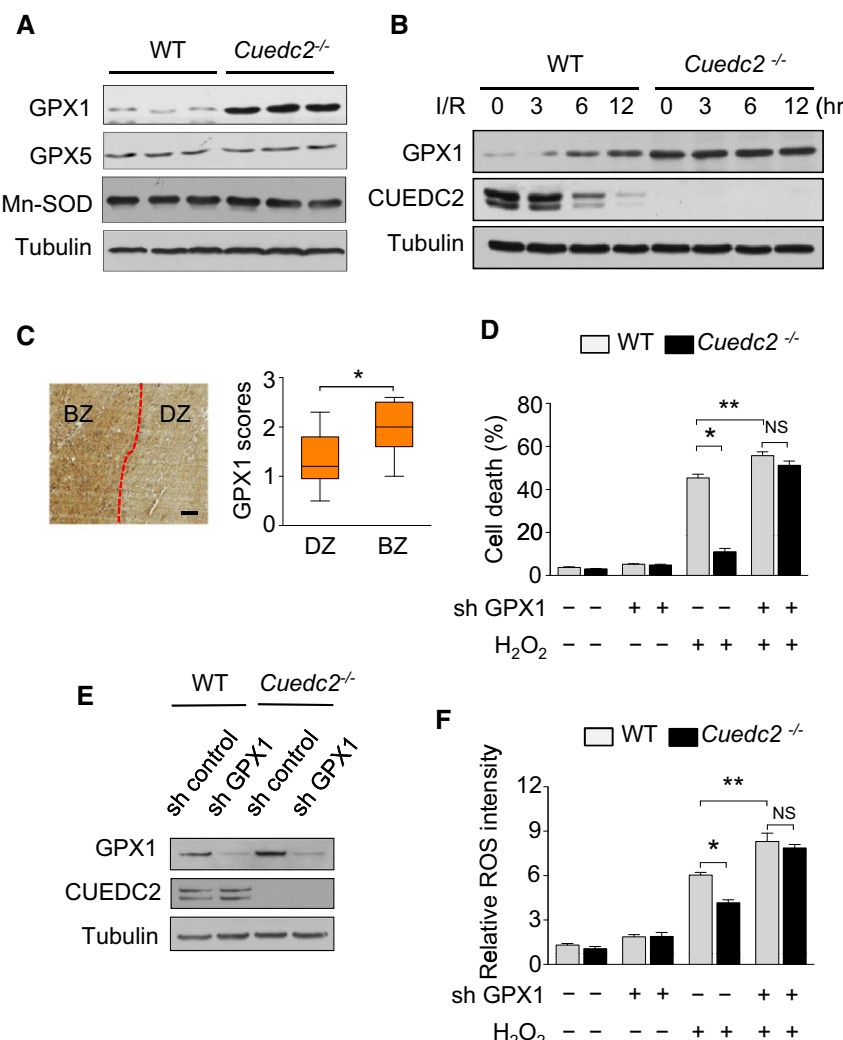

**Figure 4.   CUEDC2 deletion promotes the antioxidant potential of cardiomyocytes by upregulating GPX1 levels.**

A   Protein was extracted from the left ventricle of wild-type or $Cuedc2^{-/-}$ adult mice (12 weeks old) and subjected to immunoblot analysis (3 mice per group).

B   Wild-type or $Cuedc2^{-/-}$ mice were subjected to a heart ischemia for 30 minutes *in vivo*, followed by reperfusion for the indicated times. Protein was extracted from the area at risk of left ventricle and subjected to immunoblot analysis.

C   GPX1 protein level was plotted using the immunohistochemical scores as described in methods. (left) Representative images from immunohistochemical staining for GPX1 in tissues from patients who suffered acute myocardial infarctions. (right) Plot of GPX1 scores in each BZ and DZ (n = 9 patients). DZ, distant area; BZ, border zone. Scale bars, 250 μm (left). *P = 0.0039.

D   Primary neonatal mouse cardiomyocytes expressing control or GPX1 shRNA by adenovirus infection were treated with $H_2O_2$ (1 mM) for 3 h. Cell viability was detected by FACS. *P = 0.0001; **P = 0.0275, n = 3 wells per group. NS, not significant.

E   Primary neonatal mouse cardiomyocytes were transfected with adenovirus carrying control or GPX1 shRNA, and the efficiency of GPX1 knockdown was tested by immunoblotting.

F   Primary neonatal mouse cardiomyocytes expressing control or GPX1 shRNA by adenovirus infection were subjected to $H_2O_2$ (1 mM) for 30 minutes and then stained with 5 μM CellROX® Deep Red Reagent and analyzed by flow cytometry. *P = 0.0056; **P = 0.0218, n = 3 wells per group. NS, not significant.

Data information: Data were shown as mean ± SEM and analyzed by paired *t*-test (C) or by two-way ANOVA followed by Bonferroni's multiple comparison test (D and F). Source data are available online for this figure.

differ between $Cuedc2^{-/-}$ mice and WT littermates 24 h after transient ischemia by 30-min reversible surgical ligation of the left anterior descending coronary artery. Additionally, no difference was observed in the size of the alcian blue-stained non-ischemic area after this procedure, indicating that ligature was performed reproducibly at a similar site of the left anterior coronary artery (Fig 7A). Strikingly, compared with WT mice, the ratios of the white

triphenyltetrazolium chloride (TTC)-negative I/R injury area to LV area (IRI/LV) and IRI/AAR were significantly reduced in $Cuedc2^{-/-}$ mice (IRI/LV: 28.5 ± 1.9% in WT group vs. 16.3 ± 2.3% in $Cuedc2^{-/-}$ group, P < 0.0001 IRI/AAR: 52.5 ± 3.2% in WT group vs. 31.8 ± 2.1% in $Cuedc2^{-/-}$ group, P < 0.0001 n = 15) (Fig 7A). Consistent with this observation, we found that $Cuedc2^{-/-}$ cardiomyocytes were more resistant to I/R-induced cell death than

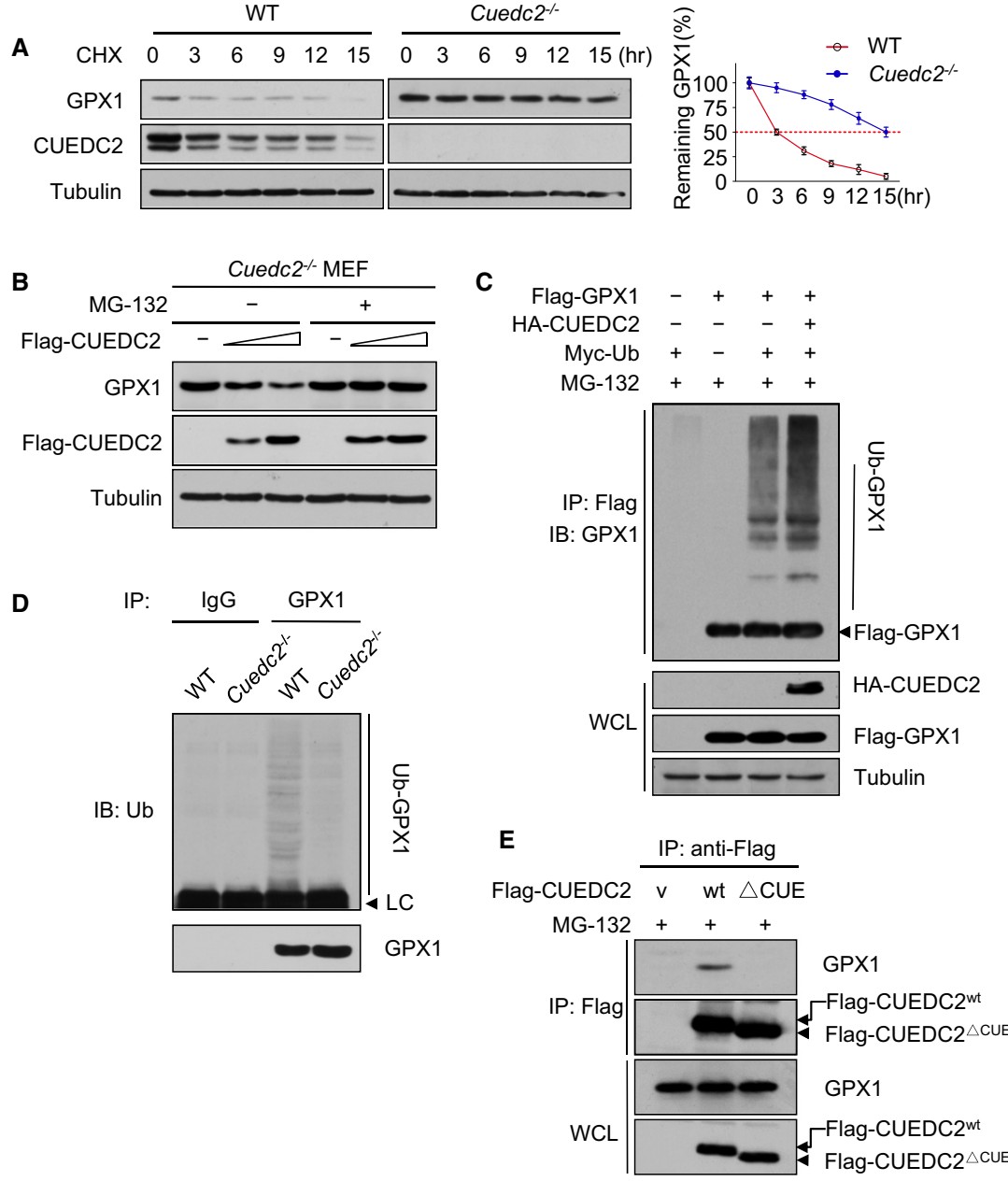

**Figure 5.  CUEDC2 destabilizes GPX1 by facilitating its ubiquitin-proteasome-dependent degradation.**

A   Lysates from MEF-WT or MEF-*Cuedc2*$^{-/-}$ cells treated with 20 μM cycloheximide (CHX) for the indicated times were subjected to immunoblotting (left). Relative GPX1 levels were quantified by densitometry (right).

B   MEF-*Cuedc2*$^{-/-}$ cells were transfected with increasing amounts of FLAG-CUEDC2 (mouse) plasmids (1 μg and 2 μg). At 24 h after transfection, the cells were treated with the proteasome inhibitor MG-132 (10 μM). The cells were cultured for additional 6 h and subjected to immunoblotting.

C   HEK293T cells were transfected with the FLAG-GPX1, HA-CUEDC2, and Myc-ubiquitin plasmids as indicated and treated with MG-132 (10 μM) for 6 h before harvest. Cell lysates were immunoprecipitated (IP) with anti-FLAG (M2). The immunoprecipitates were analyzed by Western blot using an anti-GPX1 antibody. Whole-cell lysates (WCL) were analyzed by Western blots with anti-FLAG or anti-HA antibody to determine the protein of Flag-GPX1 and HA-CUEDC2.

D   The heart tissue lysates from wild-type and *Cuedc2*$^{-/-}$ mice were incubated with IgG or anti-GPX1 antibody and then captured on protein G–Sepharose beads. The immunoprecipitates were analyzed by immunoblotting with an anti-ubiquitin antibody.

E   HEK293T cells were transfected with the Flag-CUEDC2 or Flag-CUEDC2$^{\triangle CUE}$ (deletion of the CUE domain of CUEDC2) and treated with MG-132 (10 μM) for 6 h before harvest. Cell lysates were immunoprecipitated (IP) with anti-Flag (M2). The immunoprecipitates were analyzed by Western blot using an anti-GPX1 antibody. Whole-cell lysates (WCL) were analyzed by Western blots with anti-Flag or anti-GPX1 antibody to determine the protein of Flag-CUEDC2 and GPX1. v: vector.

Source data are available online for this figure.

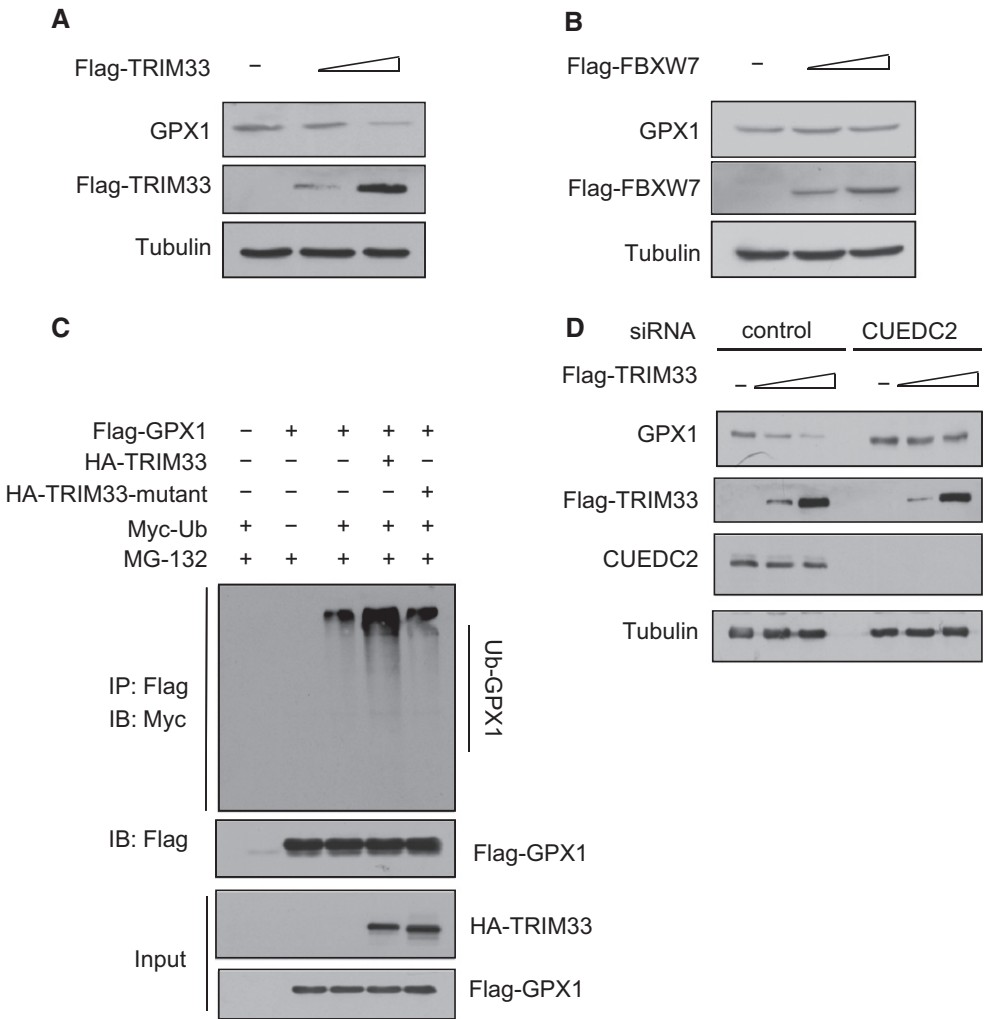

**Figure 6.  CUEDC2 facilitated E3 ligase TRIM33-mediated GPX1 degradation.**

A   HEK293T cells were transfected with increasing amounts of Flag-TRIM33 expression vectors, and 48 h after transfection, cell lysates were subjected to immunoblotting (IB) using antibodies as indicated.

B   HEK293T cells were transfected with increasing amounts of Flag-FBXW7 expression vectors, and 48 h after transfection, cell lysates were subjected to immunoblotting (IB).

C   HEK293T cells were transfected as indicated; at 18 h after transfection, the cells were treated with the proteasome inhibitor MG132(20 μM) for 6 h and then harvested. Cell lysates were immunoprecipitated by Flag antibody (M2 beads) and ubiquitin-conjugated GPX1 was detected by Western blotting with anti-Myc antibody.

D   HEK293T cells were transfected with either control or CUEDC2 siRNA and with increasing amounts of Flag-TRIM33 expression vectors; cell lysates were subjected to immunoblotting (IB) using antibodies as indicated.

Source data are available online for this figure.

WT cardiomyocytes, as assessed by TdT-mediated dUTP nick-end labeling (TUNEL)-stained nuclei, a measurement of cell apoptosis (Fig 7B and C). In addition, lactate dehydrogenase (LDH) release due to the disruption of the plasma membrane of cardiomyocytes following I/R treatment was decreased in *Cuedc2*$^{-/-}$ mice (Fig 7D). Therefore, the loss of CUEDC2 protects the cardiomyocytes from I/R-induced cell death during the early period of reperfusion. More importantly, by assessing left ventricle function with echocardiography, the cardiac function of *Cuedc2*$^{-/-}$ mice was markedly improved compared with their WT littermates 1 week following I/R (Fig 7E). In this context, though the LV ejection fraction (EF) and

LV fractional shortening (FS) were both remarkably reduced after I/R compared with the sham group, the ablation of CUEDC2 preserved the LV ejection fraction and fraction of shorting, indicating an improved cardiac function recovery. Furthermore, following I/R injury, CUEDC2 ablation led to a significant decrease in left ventricular internal dimensions in systole (LVIDs) compared with their WT littermates (Fig 7E).

We next generated a safe, efficient, and cardiotropic vector, recombinant adeno-associated viruses, serotype 9 (rAAV9) (Lin *et al*, 2014) carrying either wild-type full-length CUEDC2 (rAAV9-CUEDC2) or the control gene GFP (rAAV-GFP), and delivered these

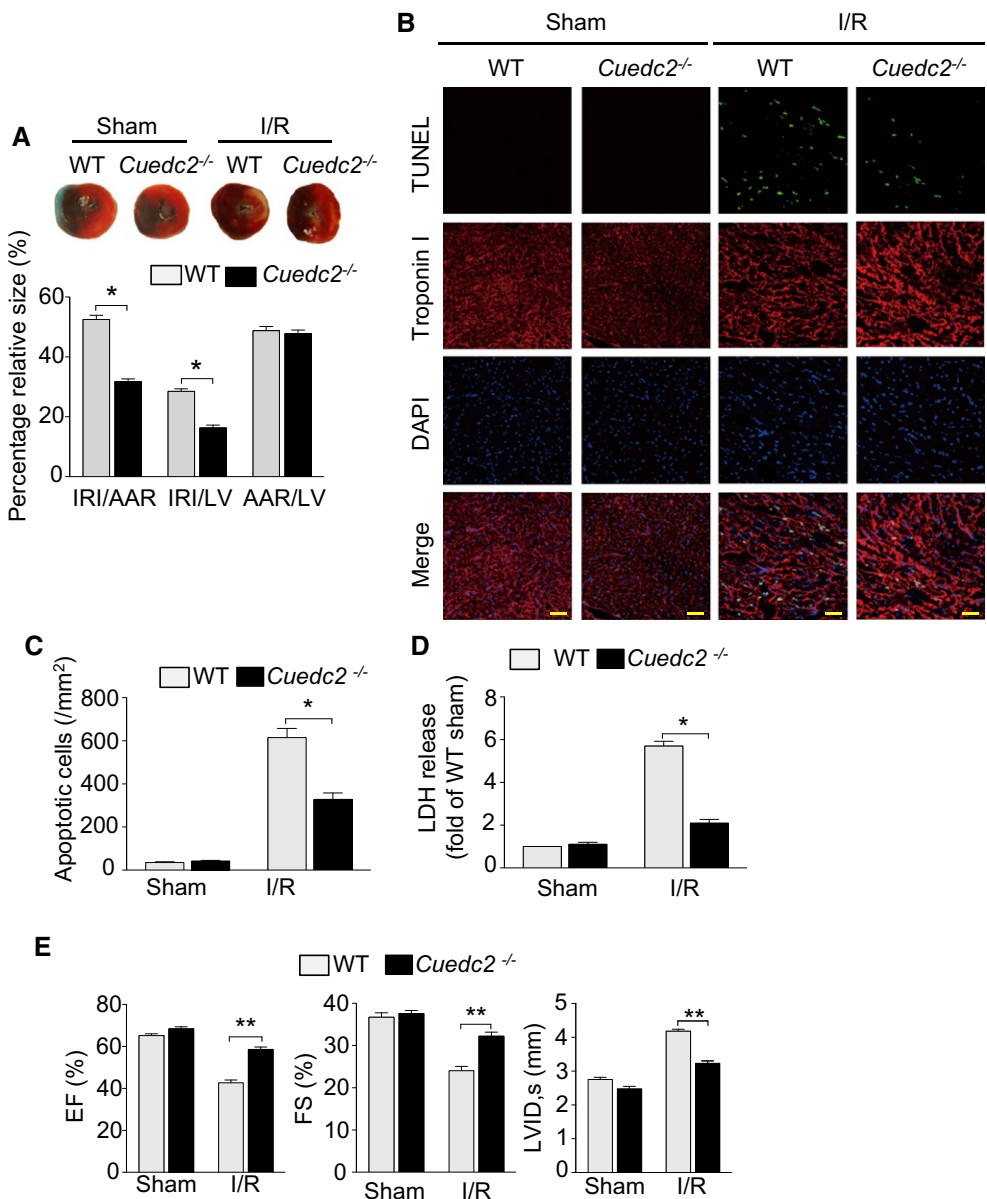

**Figure 7. CUEDC2 ablation confers the mouse heart with resistance to I/R injury.**

A   CUEDC2 ablation reduced I/R injury, which was induced by 30 minutes of ischemia followed by 24-h reperfusion. AAR, area at risk; LV, left ventricular area; IRI, area of I/R injury. The ratios of AAR to LV, IRI to AAR, and IRI to LV are shown. *P < 0.0001 compared with WT, n = 15 mice per group. (top panel) Representative cross-sectional slices derived from the hearts stained by TTC. Area of I/R injury is indicated in light pink.

B   Representative images of ventricular myocardium sections from WT and *Cuedc2⁻/⁻* mice exposed to sham operation or I/R injury (2 h after the onset of reperfusion following 30-min ischemia). Green, TUNEL-positive myocyte nuclei; red, troponin I-stained cardiomyocytes; blue, DAPI-stained nuclei. Scale bar, 50 μm.

C   Quantitative analysis of apoptosis. *P < 0.0001, n = 8 mice per group.

D   Serum LDH levels in sham-operated mice or those subjected to ischemia and a 4-h reperfusion. Data from 3 separate experiments were analyzed by one-way analysis of variance. *P < 0.0001, n = 13 mice per group.

E   Echocardiographic analysis of left ventricular dimensions and cardiac function at 1 week post-I/R in mice. **P < 0.0001, n = 15 mice in the sham-operated group and n = 10 mice in the I/R group.

Data information: Data were shown as mean ± SEM and analyzed by unpaired t-test.

genes into *Cuedc2⁻/⁻* mice. The rAAV9-CUEDC2 and rAAV-GFP gene transfer resulted in an increase in CUEDC2 and GFP protein level in the myocardium 4 weeks after gene transfer and only little expression in liver, lung, brain, and smooth muscles (Fig 8A and

Appendix Fig S12). Consistent with wild-type mice, CUEDC2 delivered into the myocardium could also be degraded after 6 h of reperfusion following 30-min ischemia. However, gene transfer of higher doses of rAAV9-CUEDC2 resulted in higher amounts of CUEDC2

remaining after 6 h of reperfusion following 30-minute ischemia. Strikingly, the size of the I/R injury area in rAAV9-CUEDC2 mice was larger than in $Cuedc2^{-/-}$ mice transferred with rAAV9-GFP (41.3 ± 7.2% in $Cuedc2^{-/-}$ + rAAV-CUEDC2 (low dose, $5 × 10^{11}$ VG) ($n = 12$) vs. 32.5 ± 6.1% in $Cuedc2^{-/-}$ + rAAV-GFP ($n = 10$), $P < 0.05$, the CUEDC2 protein level was shown in Fig 8B). In addition, the I/R injury size was much larger in $Cuedc2^{-/-}$ mice transfected with the higher dose ($1 × 10^{11}$ VG) (ratio of I/R injury area to area at risk: 46.2 ± 6.5% ($n = 12$), $P < 0.05$ compared with $Cuedc2^{-/-}$ + rAAV-GFP), though there was no significant difference between the $Cuedc2^{-/-}$ + rAAV-CUEDC2 (low dose) group and the $Cuedc2^{-/-}$ + rAAV-CUEDC2 (high dose) group (Fig 8C). Accordingly, echocardiography confirmed that heart function was worse when CUEDC2 was delivered back into the $Cuedc2^{-/-}$ heart (LV

ejection fraction: 42.6 ± 1.9% in $Cuedc2^{-/-}$ + rAAV-CUEDC2 (low dose) ($n = 12$) vs. 53.8 ± 2.5% in $Cuedc2^{-/-}$ + rAAV-GFP ($n = 10$), $P < 0.0001$. 39.8 ± 2.9% in $Cuedc2^{-/-}$ + rAAV-CUEDC2 (high dose) ($n = 12$) vs. 53.8 ± 2.5% in $Cuedc2^{-/-}$ + rAAV-GFP ($n = 10$), $P < 0.0001$) (Fig 8D). These results indicated that the ablation of CUEDC2 protected the heart from I/R injury, with significantly improved cardiac function recovery post-I/R.

### Ablation of CUEDC2 protected mouse from ageing-induced cardiomyopathy

Because oxidative stress is one of the key stimulators of ageing-induced cardiomyopathy, we next tested the overall levels of ROS in young (8-week-old) and old (20-month-old) mice. The levels of ROS

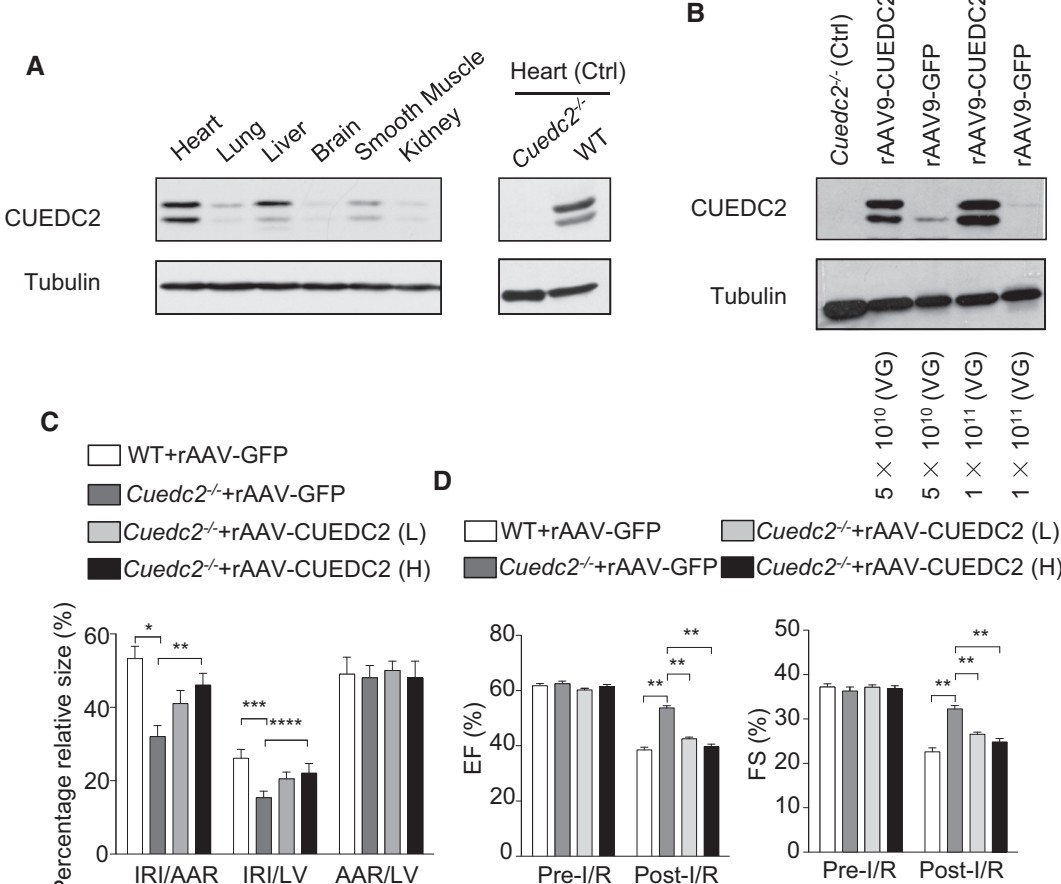

**Figure 8.  Rescuing CUEDC2 expression eliminated the protective effects in $Cuedc2^{-/-}$ mice.**

A  Immunoblotting for CUEDC2 protein level. Four weeks after tail vein injection of rAAV9-CUEDC2, indicated mice tissues were harvested and subjected to immunoblotting to detect CUEDC2 protein level. Ctl: heart of rAAV9-GFP injected mice.

B  Animals were injected by rAAV9-CUEDC2 in a different amount of viral genomes (VG). Heart tissues were then analyzed by immunoblotting to detect CUEDC2 transduction. Tubulin protein level was examined for normalization purposes.

C  The ratios of AAR to LV, IRI to AAR, and IRI to LV are shown. $n = 10$ mice in WT + rAAV-GFP group and $Cuedc2^{-/-}$ + rAAV-GFP group; $n = 12$ mice in $Cuedc2^{-/-}$ + rAAV-CUEDC2 ($1 × 10^{10}$ VG) group and $Cuedc2^{-/-}$ + rAAV-CUEDC2 ($1 × 10^{11}$ VG) group. *$P = 0.0055$, **$P = 0.0146$, ***$P = 0.0200$, ****$P = 0.0377$.

D  Effect of rAAV9-CUEDC2 treatment on echocardiography indices of left ventricle function recovery post-I/R measured by fractional shortening (FS) and ejection fraction (EF). WT+ rAAV-GFP ($n = 10$); $Cuedc2^{-/-}$ + rAAV-GFP ($n = 10$); $Cuedc2^{-/-}$ + rAAV-CUEDC2 (Low dose) ($n = 12$); $Cuedc2^{-/-}$ + rAAV-CUEDC2 (High dose) ($n = 12$). **$P < 0.0001$.

Data information: Data were shown as mean ± SEM and analyzed by two-way ANOVA followed by Bonferroni's multiple comparison test.
Source data are available online for this figure.

    

were comparable in the hearts of young *Cuedc2*$^{-/-}$ and WT mice. The cardiac ROS levels were increased dramatically in the hearts of old WT mice while detecting DHE oxidation product with high-performance liquid chromatography (HPLC) (Fig 9A and B); however, this ROS upregulation was considerably suppressed in the hearts of old *Cuedc2*$^{-/-}$ mice (Fig 9A and B), indicating that the ablation of CUEDC2 has little effect on the basal physiological ROS levels of the young heart, but specifically inhibits the detrimental increase in ROS levels in the ageing mouse heart. Then, we tested the levels of 8-oxo-2′-deoxyguanosine (8-oxo-dG) in young and old mice. Interestingly, we found that 8-oxo-dG was increased in WT old mice, but to a significantly less degree in *Cuedc2*$^{-/-}$ old mice (Fig 9C). These results indicated that GPX1 upregulation during CUEDC2 ablation protects the heart from ageing-associated oxidative stress injury. Then, we measured the key features of ageing-induced cardiomyopathy including HW/BW ratio, myocyte size, LVFS, LVEF, fibrosis, and apoptosis. HW/BW ratio, an index of cardiac hypertrophy (CH), was not different between young WT and *Cuedc2*$^{-/-}$ mice and was significantly elevated in old WT mice, but not in old *Cuedc2*$^{-/-}$ mice (Fig 9D). However, there is no difference in the body weights of the WT and *Cuedc2*$^{-/-}$ mice (Appendix Fig S13). Similarly, the development of myocyte hypertrophy was significantly attenuated in the old *Cuedc2*$^{-/-}$ mice compared with control mice, confirming that less CH developed in old *Cuedc2*$^{-/-}$ mice. Importantly, the ablation of CUEDC2 had beneficial effects on cardiac function in aged mice because LVFS and LVEF were significantly higher in old *Cuedc2*$^{-/-}$ mice compared with control mice (Fig 9E). Ageing cardiomyopathy is also characterized by increased apoptosis, which was observed in the present study in the old WT mice, but much less in the *Cuedc2*$^{-/-}$ mice (Fig 9F). Myocardial fibrosis, an important index of detrimental remodeling, was also increased in WT old mice vs WT young mice, and this was significantly attenuated in the old *Cuedc2*$^{-/-}$ mice (Fig 9G and H). Together, these results indicate that the ablation of CUEDC2 mitigates ageing-induced cardiomyopathy by suppressing the abnormal accumulation of ROS in the myocardium.

## Discussion

Loss of cardiomyocytes during I/R injury triggers myocardial remodeling, leading to hypertrophy of surviving cardiomyocytes, cardiac interstitial fibrosis, and eventually heart dysfunction (Qian *et al*, 2011). Because cardiomyocytes are terminally differentiated and have little potential for proliferation, inhibiting cardiomyocyte death remains one of the most attractive strategies for the treatment of heart disease. Although extensive studies have been devoted to develop such strategies, there have been few successes in the clinics (Hausenloy *et al*, 2007). Here, we discovered that CUEDC2 was a novel and critical component involved in the regulation of intrinsic antioxidant defense. Our findings indicate that the ablation of CUEDC2 promotes cardiomyocyte survival by suppressing ROS generation and the associated cell death pathways, thereby improving heart function recovery after ischemic injury. Moreover, the ablation of CUEDC2 had no significant effect on the basal levels of cardiac ROS, which is required for the normal physiological functions of cardiomyocytes. In addition, the ablation of CUEDC2 *in vivo* did not perturb normal heart development and function. Therefore,

CUEDC2 represents an ideal therapeutic target to treat MI and ageing-induced cardiomyopathy.

In the heart, redox signaling is involved not only in development and physiology but also in pathological processes (Burgoyne *et al*, 2012). Basal ROS plays an important role in cardiomyocyte differentiation, proliferation, excitation–contraction coupling, and the regulation of heart blood flow (Burgoyne *et al*, 2012). Conversely, ROS acts in some instances as second messengers downstream of specific ligands, including transforming growth factor-β1 (TGF-β1) and endothelin, and is involved in modulating the activity of specific transcription factors, including NF-κB and activator protein-1 (AP-1) (Tsung *et al*, 2005). Therefore, abnormally high levels of ROS play critical roles in I/R injury, myocardial remodeling, cardiac inflammation, heart failure, and ageing-induced cardiomyopathy (Giordano, 2005). In our study, we found that the ablation of CUEDC2 had no significant effect on basic ROS but specifically enhanced ROS scavenging under conditions of stress. Therefore, the ablation of CUEDC2 inhibited cardiomyocyte death and improved heart function under oxidative stress.

It is well established that GPX1 protects the heart from ischemic injury by promoting ROS scavenging (Negoro *et al*, 2001). Previous studies have shown that GPX1 expression can be modulated at multiple stages, that is, at the mRNA level, at the post-transcriptional level, and through the translational modifications. Several other reports have indicated that GPX1 activity may be modulated by post-translational modifications, including selenocysteine oxidation (Cho *et al*, 2010) and phosphorylation (Duval *et al*, 2002). We showed that the protein levels of GPX1 could be modulated by TRIM33-dependent degradation, a process that was facilitated by CUEDC2. CUEDC2 protein contains a CUE domain (coupling of ubiquitin conjugation to endoplasmic reticulum degradation domain). It was reported that CUE is a kind of ubiquitin-binding domains (UBDs), which bind multiple ubiquitin molecules and promote the ubiquitination. As the CUE domain was indispensable for the regulation of CUEDC2 on GPX1, CUEDC2 might facilitate TRIM33-mediated GPX1 degradation by binding multiple ubiquitin molecules. This finding identified a novel mechanism of regulating cellular GPX1 protein level. In view of the rapid degradation of CUEDC2, accompanied by increased levels of GPX1 following I/R injury, our findings provide a novel intrinsic mechanism to effectively salvage cardiomyocytes within the ischemic border zone (Appendix Fig S14).

In previous studies, a variety of other downstream effectors of CUEDC2 have been identified. To test whether the effects observed here are mediated in part through the previously described activities of CUEDC2, we examined NF-κB activity and the levels of the progesterone receptor (PR) and estrogen receptor-α (ER-α) in the hearts of WT and *Cuedc2*$^{-/-}$ mice (Appendix Fig S15). Because we have previously reported that CUEDC2 was a negative regulator of NF-κB (Li *et al*, 2008), NF-κB activity in the heart was expected to be upregulated when CUEDC2 was ablated. However, in our present study, I/R-induced NF-κB activation was dramatically reduced in the hearts of *Cuedc2*$^{-/-}$ mice when compared to the hearts of WT mice. In our previous study, we found that CUEDC2 blocks NF-κB activation through inhibiting IKK activity and that CUEDC2 specifically blocks TNF-α- or IL-1-induced, but not UV-triggered, NF-κB activation (Zhang *et al*, 2013), indicating that the regulation of NF-κB activation by CUEDC2 is signal pathway dependent. Because it is well known that ROS affects the downstream events of IKK

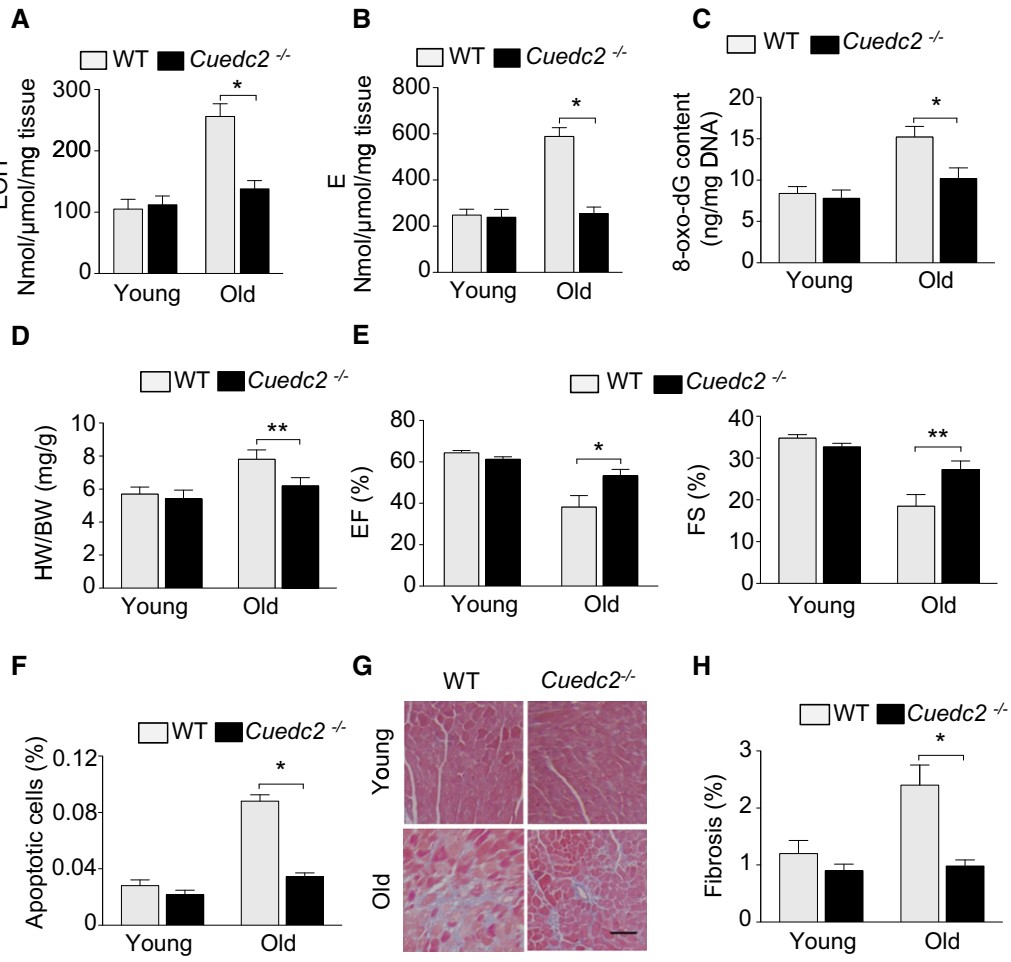

**Figure 9.** *Cuedc2*$^{-/-}$ mice are protected from ageing-induced cardiomyopathy.

A     HPLC detection of a superoxide probe oxidized dihydroethidium (DHE) product in young (8-week-old) and old (20-month-old) mouse heart tissue, 2-hydroxyethidium (EOH), a specific product for superoxide anion radical. *$P$ = 0.0087, $n$ = 5 in each group.

B     HPLC detection of ethidium (E), oxidized by other reactive oxygen species such as $H_2O_2$ (mainly) and ONOO. *$P$ = 0.0021, $n$ = 5 in each group.

C     Comparison of 8-hydroxy-2′-deoxyguanosine (8-OHdG) in young and old mice. *$P$ = 0.0164, $n$ = 6 mice per group.

D     Comparison of heart weight/body weight (HW/BW) in WT and *Cuedc2*$^{-/-}$ mice. **$P$ = 0.036, $n$ = 15 in young WT and *Cuedc2*$^{-/-}$ mice, $n$ = 10 in old WT mice, and $n$ = 12 in old *Cuedc2*$^{-/-}$ mice.

E     Comparison of the left ventricle function in WT and *Cuedc2*$^{-/-}$ mice by transthoracic echocardiography. *$P$ = 0.0384, **$P$ = 0.0389. $n$ = 15 in young WT and *Cuedc2*$^{-/-}$ mice, $n$ = 10 in old WT mice, and $n$ = 12 in old *Cuedc2*$^{-/-}$ mice.

F     Left ventricle cardiomyocyte apoptosis was tested by TUNEL staining. *$P$ = 0.0017, $n$ = 5 in each group.

G, H   Left ventricle myocardial fibrosis was tested by Masson staining in WT and *Cuedc2*$^{-/-}$ mice. The represented results were shown in (G) and the quantitative analysis of myocardial fibrosis was shown in H. *$P$ = 0.0177, $n$ = 5 in each group.

Data information: Data were shown as mean ± SEM and analyzed by unpaired *t*-test.

activation in the canonical NF-κB pathway (Li *et al*, 2008), the decrease in I/R-induced NF-κB activation could be due to the lower overall ROS levels in the *Cuedc2*$^{-/-}$ heart. We have also reported that CUEDC2 promotes the ubiquitination and degradation of ER-α and PR in breast cancer cells. In this study, we examined the protein levels of ER-α and PR in the heart and found that there was no difference in the levels of these proteins in WT and *Cuedc2*$^{-/-}$ hearts. Taken together, these results suggest that the previously described activities of CUEDC2 do not mediate the effect observed in heart. Instead, CUEDC2 targets the antioxidant enzyme GPX1 to regulate cardiac antioxidant defenses. It is worthy to note that *Cuedc2* gene

was globally knocked out in mice in our study. Our findings suggest the CUEDC2 loss in the cardiomyocytes is an important mechanism of protecting the heart from I/R injury. Nevertheless, we could not fully rule out the possibility that the loss of CUEDC2 in other cell types might contribute to the heart protection as well. Given that CUEDC2 plays various roles under different conditions, the heart-specific delivery should be taken into account to avoid the potential side effects in other organs if developing a therapeutic strategy to inhibit CUEDC2 for I/R injury protection.

Following I/R, CUEDC2 protein level gradually decreased, which led to the increase in GPX1 protein level to scavenge ROS.

Therefore, CUEDC2 degradation upon I/R is an intrinsic protective mechanism against I/R injury in the heart. Previously, we reported that CUEDC2 protein could be degraded under UV treatment, and phosphorylation of CUEDC2 facilitates this degradation (Zhang *et al*, 2013). In this study, the decrease in CUEDC2 in response to ischemia/reperfusion is not at mRNA level. We postulated that the post-translational modification might regulate the stability of CUEDC2. We have identified some E3 ligase candidates for CUEDC2 degradation by immunoprecipitation and mass spectrometry analysis. The potential mechanism of the E3 ligase responsible for CUEDC2 destruction following I/R injury is being investigated now. This is of high significance for developing a potential therapeutic strategy to reduce CUEDC2 levels in the context of myocardial injury.

In conclusion, our findings discover a critical component involved in redox homeostasis in response to oxidative stress. Considering the widespread expression of both CUEDC2 and GPX1, this novel pathway could play important roles in regulating the ROS levels in other physiological systems. ROS is a primary inducer of many ageing-related human diseases such as neurodegeneration and cancer, which are also associated with the loss of GPX1 activity (Lubos *et al*, 2011). Therefore, in addition to treating reperfusion injury that occurs following MI, percutaneous coronary intervention, or coronary bypass surgery, CUEDC2 represents a potential therapeutic target to treat ageing-related human diseases.

## Materials and Methods

### Human heart samples

Human heart samples were obtained from BioCat GmbH (Heidelberg, Germany). The BioCat GmbH stated that all tissues were collected with the donor being informed completely and with their consent. The BioCat GmbH makes sure that they follow standard medical care and protect the donors' privacy. Samples of infarct hearts were collected from the left ventricles of individuals with ischemic myocardial infarction. Non-ischemic hearts (which were used as normal controls) were obtained from donors who died from neurological diseases or motor vehicle accidents. The information of clinical samples is listed in Appendix Table S1. Due to the strict regulations to preserve donors' anonymity in Germany, more information about these patients was not available. The infarcted area was defined based on the morphological changes in the heart tissue, that is, sporadic and local necrotic area accompanied by neutrophil infiltration, cellular edema, and nuclear changes. The area around the infarcted area with normal morphological cardiomyocytes was defined as border zone (BZ). CUEDC2 protein level was plotted using the immunohistochemical scores as described in the previous study (Pan *et al*, 2011). All the experiments based on this human heart samples were confirmed to the principles set out in the WMA Declaration of Helsinki and the Department of Health and Human Services Belmont report.

### Mouse *in vivo* cardiac I/R model

For all animal studies, institutional approval by Institutional Animal Care and Use Committee of the National Center of Biomedical

Analysis was granted. Experiments were performed in compliance with ARRIVE guidelines. WT and *Cuedc2*$^{-/-}$ mice were housed in individual, ventilated cages (IVCs) with 12-h light/dark cycles with food and water *ad libitum*. All mouse lines used for further experiments were established on a C57BL/6 background as we reported previously (Chen *et al*, 2014). All surgeries and subsequent analyses were performed in a blinded fashion for genotype and intervention. C57BL/6 male, WT, and *Cuedc2*$^{-/-}$ mice (8–12 weeks old, body weight 18–25 g) were obtained from our mouse facility and were anesthetized with 2.4% isoflurane and 97.6% oxygen and placed in a supine position on a heating pad (37°C). Mice were randomly assigned to sham group or I/R group. Animals were intubated with a 19-gauge stump needle and ventilate with room air (Harvard Rodent Ventilator; Harvard; SV, 250 μl; respiratory rate, 110 breaths per min). Cardiac I/R was induced by 30-min reversible surgical ligation of the left anterior descending coronary artery with a 7-0 Prolene suture, then reperfused for different times as indicated in figures. Sham-operated animals served as surgical controls and were subjected to the same procedures as the experimental animals with the exception that the left anterior descending artery was not ligated. In the analysis of heart tissue in the animal experiments, the samples were from the area at risk following I/R treatment. This area includes the infarct and border zone. After I/R treatment, the left ventricle was divided into the area at risk and normal area. Myocardial infarction only occurred in the area at risk. Therefore, we analyzed the area at risk for the differences between WT and *Cuedc2*$^{-/-}$ mice to assess the role of CUEDC2 in protecting the heart from I/R injury.

### Determination of the AAR and myocardial I-R injury

Twenty-four hours after reperfusion, the mice were anesthetized and cannulated with tubing, and left anterior descending (LAD) coronary artery was ligated again; 2% alcian blue (Sigma Aldrich) was perfused into the aorta; thus, all myocardial tissues were stained blue except the AAR. The LV was isolated and cut into four or five pieces of about 1 mm with the first cut at the ligation level. LV slices were stained in 1.5% TTC for 30 min at 37°C and then fixed in 4% PFA overnight at 4°C. The area of myocardial infarct (MI) was demarcated as a white area, whereas viable myocardium was stained red. Photographs were taken for both sides of each section. The AAR and the MI were determined via planimetry by using the computer software ImageJ (http://imagej.nih.gov/ij/). Infarct size was calculated as the percentage of MI compared with the AAR. For the sham group, the ligature was briefly placed on the LAD during alcian blue injection in order to define the area at risk, without 30-min ischemia period.

### Transthoracic echocardiography in mouse

Echocardiography was performed by the Vevo 770 High-Resolution Micro-Imaging System (VisualSonics) with a 15-MHz linear array ultrasound transducer. The LV was assessed in both parasternal long- and short-axis views at a frame rate of 120 Hz. End systole or end diastole was defined as the phase in which the smallest or largest area of LV, respectively, was obtained. Left ventricular end-systolic diameter and left ventricular end-diastolic diameter were

measured from the LV M-mode tracing with a sweep speed of 50 mm s$^{-1}$ at the papillary muscle level. EF, ejection fraction, was calculated as $[(EDV - ESV)/EDV] \times 100$, where EDV represents end-diastolic volume and ESV end-systolic volume. FS, fractional shortening of left ventricular diameter, was calculated as $[(LVIDd - LVIDs)/LVIDd] \times 100$, where LVIDd represents diastolic left ventricular internal diameters and LVIDs systolic left ventricular internal diameters.

**Primary cardiomyocyte isolation and culture**

Neonatal cardiomyocytes were isolated from wild-type and *Cuedc2*$^{-/-}$ postnatal day 3 mice as reported previously with little modification. In brief, ventricles were washed and minced after dissection in HEPES-buffered saline solution containing 130 mM NaCl, 3 mM KCl, 1 mM NaH$_2$PO$_4$, 4 mM glucose, and 20 mM HEPES. We dispersed the tissues in a series of incubations at 37°C in HEPES-buffered saline solution containing 0.15% trypsin (Sigma) and 200 U ml$^{-1}$ collagenase II (Invitrogen). After centrifugation, the cells were resuspended in DMEM/F-12 medium (GIBCO) containing 5% (vol/vol) horse serum, 0.1 mM ascorbate, insulin-transferring sodium selenite media supplement, 100 U ml$^{-1}$ penicillin, and 100 μg ml$^{-1}$ streptomycin. We preplated the dissociated cells at 37°C for 1 h, diluted them to $1 \times 10^6$ cells ml$^{-1}$, and plated them in culture dishes (Corning). For hypoxia treatment, primary mouse cardiomyocytes were cultured in serum-free media under hypoxic conditions (0.3% O$_2$, 37°C) by using a GasPak Plus system (Becton-Dickinson).

**Cardiomyocyte viability assays**

Cardiomyocyte viability was detected by flow cytometry using a LSRII instrument (Becton-Dickinson) as described previously after staining with 50 μg ml$^{-1}$ propidium iodide (PI, Invitrogen). The cells were gently mixed and incubated at room temperature for 15 min.

**Determination of myocardial injury by LDH release**

Serum was obtained from post-reperfusion mouse, and LDH was spectrophotometrically assayed with the kit from Sigma Chemical Co (St Louis, MO).

**Cellular reactive oxygen species (ROS) measurement**

Primary mouse cardiomyocytes were seeded at $5 \times 10^5$ cells/well (24-well) tissue culture plates. Cells were labeled with 5 μM of CellROX Deep Red (Invitrogen) for 30 min (37°C, 5% CO$_2$), washed twice and then stimulated as indicated in the figure legends. The cells were then harvested and analyzed by flow cytometry. Viable cells (PI negative) were quantified for ROS production by calculating the mean fluorescence intensity (MFI) of at least 5,000 events with Weasel V2.0 analysis software.

**HPLC detection of tissue DHE oxidation products**

Thirty minutes after the onset of reperfusion, the area at risk of mouse hearts was harvested and cut into 3 pieces and weighed.

Each piece was immediately incubated with DHE (Life Technologies, D-23806, 100 μM in PBS) at 37°C for 30 min. The buffer was removed and the sample washed once with PBS. DHE and oxidized products were extracted with acetonitrile (500 μl) and briefly sonicated ($3 \times 30$ s, 8 W). Samples were spun down (12,000 *g*, 10 min at 4°C), and the supernatant was collected and dried under vacuum. The samples were further dissolved in 120 μl PBS–DTPA and injected into the HPLC system as described previously (Puente *et al*, 2014).

**rAAV vector production and purification**

rAAV9-CUEDC2 and rAAV9-GFP were produced using the two-plasmids protocol as previously described with the following modifications: HEK293T cells (ATCC) were grown in triple flasks for 24 h (DMEM, 10% fetal bovine serum) before adding the calcium phosphate precipitate. After 72 h, the virus was purified from benzonase-treated crude cell lysates over an iodixanol density gradient (Optiprep, Greiner Bio-One Inc.), followed by heparin–agarose type I affinity chromatography (Sigma). Finally, viruses were concentrated and formulated into lactated Ringer's solution (Baxter Healthcare Corporation) using Vivaspin 20 centrifugal concentrators 50K MWCO (Vivascience Inc.) and stored at −80°C.

**Western blotting**

For the heart tissue, the total protein was lysated from left ventricle or the area at risk post-I/R treatment. Antibodies used in these studies were as follows: rabbit polyclonal anti-CUEDC2 antibody (Cat. No. 4839) and rabbit polyclonal anti-CUEDC1 antibody (Cat. No. 4829) were obtained from ProSci Incorporated. Rabbit polyclonal anti-GPX1 (Cat. No. 2971) was obtained from Epitomics; rabbit polyclonal anti-GPX5 (sc390093) antibody was obtained from Santa Cruz Biotechnology. And anti-MnSOD antibody ((ab13533)) was obtained from Abcam. All the primary antibodies were diluted 1,000 times when used for immunoblotting.

**Luciferase reporter assays**

Wild-type or *Cuedc2*$^{-/-}$ MEFs were transiently transfected with 0.2 mg of the luciferase reporter pNF-κB-Luc, plus 0.02 μg of the *Renilla* reporter pRL-TK. After being treated for 6 h with 10 ng/ml of TNF (Sigma) or hypoxia/reoxygenation, transfected cells were collected. Luciferase activity was assessed using a Dual-Luciferase reporter Assay system (Promega). Penilla luciferase activity was used as an internal control for transfection. All experiments were repeated at least three times.

**Ubiquitination assay**

HEK293T cells were transfected for 24 h, and MG-132 was added for another 6 h. Total cell lysates were prepared in lysis buffer containing 20 mM Tris–HCl, pH 7.5, 150 mM NaCl, 10 mM EDTA, 1% (vol/vol) Triton X-100, 1% (vol/vol) deoxycholate, and protease inhibitor cocktail (04-693-132-001; Roche). Cells was lysed using sonication and centrifuged. Then, the supernatants were immunoprecipitated 4 h with constant mixing at 4°C with mouse anti-FLAG-M2 beads (A2220, Sigma). The immunocomplexes

were washed 5 times and subjected to SDS–PAGE followed by immunoblotting with anti-Myc (Santa Cruz) and anti-GPX1 (Epitomics).

The hearts were isolated from wild-type and $Cuedc2^{-/-}$ mice separately. Then, the heart tissues were homogenized, lysed with RIPA buffer (containing 50 mM Tris–HCl, PH 7.4, 150 mM NaCl, 1% NP-40, 0.5% sodium deoxycholate, 0.1% SDS, 1 mM PMSF, 20 mM β-glycerol phosphate, 0.1 mM $Na_3VO_4$, 20 mM PNPP, 10 mM MG132, 100 μg ml$^{-1}$ leupeptin, and protease inhibitors), and sonicated for 5 min. The samples were centrifuged at 15,000 g for 4 min at 4°C. The each 0.5 mg total protein was incubated with 6 μg mouse IgG or anti-GPX1 antibody (Abnova) for 4 hrs at 4°C, then captured on protein G–Sepharose beads (GE Healthcare) for 2 hrs at 4°C. The immunoprecipitates were analyzed by immunoblotting with anti-ubiquitin antibody.

### SDS–PAGE and in-gel tryptic digestion

Freshly frozen hearts were minced and homogenized in 1.0 ml of ice-cold homogenization buffer (25 mM MOPS, 1.0 mM EDTA, pH 7.4) using a Polytron homogenizer. Aliquots of the homogenates containing 60 mg total protein were mixed with SDS and an internal standard containing 8 pmol bovine serum albumin. The samples were mixed well and heated at 70°C to assure the complete dissolution before desalting by precipitation in 1 ml of acetone overnight at −20°C. The protein pellet was solubilized in 60 ml Laemmli sample buffer and 20 mg protein was loaded in a 12.5% SDS–PAGE gel. The gel was run for approximately 15 min at 150 V to give a 1.5-cm gel. The gel was fixed, washed with several changes of water, and stained for 5 min with Coomassie Blue. Each lane was cut as a single sample and the gel piece divided roughly into 8–10 pieces. The gel pieces were destained in 50% ethanol, 40% water, and 10% acetic acid overnight at 50°C with several changes as needed for complete destaining. A standard in-gel digestion method was used. The reduction and alkylation reagents were removed and digestion was carried out by adding 1 mg trypsin (Promega) in 200 ml 10 mM ammonium bicarbonate for overnight at room temperature. The peptides produced were collected by extraction in 200 ml 50% ethanol, 50% water with 1% formic acid. The extract was evaporated to dryness and reconstituted in 150 ml 1% acetic acid in water for LC–tandem MS analysis.

### Affinity purification of GPX1 binding proteins

Flag-tagged GPX1 transiently expressed in HEK293T cells was purified with anti-FLAG-M2 beads (A2220, Sigma) in Flag-tag protein purification buffer (50 mM HEPES [pH 7.5], 150 mM NaCl, 1 mM dithiothreitol, 0.05% NP-40, and 1 mM phenylmethanesulfonyl fluoride) according to the manufacturer's instruction. The bead-bound proteins were eluted by boiling for 3 min with loading buffer and then subjected to SDS–PAGE. The proteins were detected with Coomassie Blue staining, and specific bands were excised and analyzed by LC–tandem MS analysis.

### Liquid chromatography–tandem mass spectrometry

About 10 ml of the samples was injected and loaded onto the column at 2 ml min$^{-1}$ with 0.1% formic acid. The column was

eluted at 160 nl min$^{-1}$ with a linear gradient of CH3CN in water with 0.1% formic acid (3% $CH_3CN$ to 63% $CH_3CN$ in 30 min). The triple quadrupole mass spectrometer was operated in the selected reaction-monitoring (SRM) mode. SRM conditions were managed through the program Pinpoint (Thermo Scientific) and included 2–3 peptides from each protein with 6–8 fragmentation reactions per peptide. Scheduling was used to monitor each peptide in a 4-min time window centered on the elution time of the peptide. Integrated chromatographic peak areas for each peptide were determined using the Pinpoint program. The response for each protein was calculated as the total integrated area for all peptides monitored for that protein. Data were analyzed as the raw total integrated area and after normalization to the internal standard protein.

### Real-time PCR assays

Total RNA isolation and real-time PCR assays were performed as previously reported (Chen et al, 2014). Relative quantification of gene expression level was performed using the $^{\Delta\Delta}Ct$ method with normalization to GAPDH, using the data analysis module (Roche). We used the primers listed in Appendix Table 3.

### Electrophoretic mobility shift assay (EMSA)

Preparation of tissue nuclear extracts and EMSAs were performed as described previously. Biotin-labeled NF-κB probe (AGTTGAGGG-GACTTTCCCAGGC) was purchased from Beyotime (Nantong, China); 10 μg of nuclear proteins of each specimen was incubated with 50 fmol of the biotin-labeled probe for 30 min at room temperature in the presence of 1 μg of poly (dI:dC). Protein–DNA complexes were separated from the free DNA probe by electrophoresis via a 6.5% native polyacrylamide gel. The gel was run in the 0.5 × Tris–borate–ethylene diamine tetraacetic acid (TBE) at room temperature with 30 mA for 2.5 h. The separated proteins were transferred to PVDF membranes (EMD Millipore) at 380 mA for 30 min, cross-linked under the ultraviolet lamp for 10 min, and subjected to gentle shaking in sealing liquids containing streptavidin–HRP conjugate for 15 min. The membrane was washed and balanced, followed by the addition of 5 ml BeyoECL plus Reagent A and 5 ml BeyoECL plus Reagent B. Proteins were finally analyzed with chemiDoc™ MP Imaging System (Bio-RAD). Competition tests used a 50-fold excess of the unlabeled oligoduplex.

### Statistics

Statistical analysis was performed using the software GraphPad Prism 6 (GraphPad Software, San Diego, CA). All data were analyzed for normality by Kolmogorov–Smirnov test before applying parametric statistical tests. Datasets were analyzed either by Student's $t$-test or by two-way ANOVA followed by Bonferroni's multiple comparison test, as indicated in the figure legends. Based on data from previous projects or from preliminary experiments, we calculated the sample size using G*Power 3.1.9.2 to ensure the adequate power of key experiments in detecting prespecified effect sizes. $P$-values < 0.05 were considered significant. The exact (Student's $t$-test) or adjusted (ANOVA) $P$-values for all experiments with indicated statistical significance were reported in the figure legends.

**The paper explained**

**Problem**
The irreversible loss of cardiomyocytes due to oxidative stress is the main cause of heart dysfunction following ischemia/reperfusion (I/R) injury and ageing-induced cardiomyopathy. The intrinsic antioxidant defense under ischemia/reperfusion is not fully understood.

**Results**
We found that CUEDC2, which was highly expressed in the heart, was decreased following I/R injury. CUEDC2 increased E3 ligase TRIM33-mediated GPX1 ubiquitination and proteasome-dependent degradation. Ablation of CUEDC2 enhanced the antioxidant capacity of cardiomyocytes and protected the heart from ischemia/reperfusion injury.

**Impact**
Our findings suggest a novel role of CUEDC2 in intrinsic antioxidant defense under ischemia/reperfusion. Manipulating CUEDC2 level might be an attractive therapeutic strategy for promoting cardiomyocyte survival following oxidative stress-induced cardiac injury.

Expanded View for this article is available online.

## Acknowledgements

We thank Z.G. Liu (National Cancer Institute, National Institutes of Health) for his critical review of this work. This work was supported by the National Basic Research Program of China (2014CB910603 and 2012CB910801), National Natural Science Foundation of China (no. 31370915; no. 81171917; no. 81221004; no. 81130037; no. 81270228; no. 81325014; no. 81370004), Beijing Municipal Science and Technology Commission (2012041), National Major Scientific and Technological Special Project for "Significant New Drugs Development".

## Author contributions

ZJ, A-LL, and Y-BX contributed to all studies; BL contributed to the molecular study; XP participated in data analysis and paper preparation; GX, S-SG, and TL contributed to ubiquitination and protein interactions; TZ contributed to the preparation of antibodies and data analysis.

## Conflict of interest

The authors declare that they have no conflict of interest.

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
