## [Review Process File · EMBO Molecular Medicine]

CUEDC2 modulates cardiomyocyte oxidative capacity by regulating GPX1 stability

Zhao Jian, Bing Liang, Xin Pan, Guang Xu, Sai-Sai Guo, Ting Li, Tao Zhou, Ying-Bin Xiao, and Ai-Ling Li

Corresponding author: Ying-Bin Xiao, Institute of Cardiovascular Surgery, Xinqiao Hospital, Third Military Medical University

Review timeline:

Submission date:	28 October 2015
Editorial Decision:	27 November 2015
Revision received:	11 March 2016
Editorial Decision:	26 April 2016
Revision received:	12 May 2016
Accepted:	13 May 2016

Transaction Report:

Editor: Céline Carret

1st Editorial Decision

27 November 2015

Thank you for the submission of your manuscript to EMBO Molecular Medicine. We have now heard back from the three referees whom we asked to evaluate your manuscript. As you will see from the reports below, the referees find the topic of your study of potential interest. However, they raise substantial concerns on your work, which should be convincingly addressed in a major revision of the present manuscript.

Although the referees appreciate the topic of the work, they also raised a number of concerns about the conclusiveness of the results and several technical issues that preclude appropriate evaluation of the findings. Of particular interest to us are the issues highlighted regarding the human tissue data, preparation of primary cardiac cells not explained, I/R cell culture experiments that need strengthening, further molecular understanding of what is going on as CUEDC2 has been implicated in other mechanisms under other contexts, make the co-IP data a main figure, explain discrepancies regarding the echocardiography on anaesthetised mice, and finally, a model-figure to summarise the findings would help the readers considerably.

Overall it is clear that publication of the manuscript cannot be considered at this stage. I also note that addressing the reviewers concerns in full will be necessary for further considering the manuscript in our journal and this appears to require a lot of additional work and experimentation. I am unsure whether you will be able or willing to address those and return a revised manuscript within the 3 months deadline. On the other hand, given the potential interest of the findings, I would be willing to consider a revised manuscript with the understanding that the referee concerns must be

fully addressed and that acceptance of the manuscript would entail a second round of review. I should remind you that it is EMBO Molecular Medicine policy to allow a single round of revision only and that, therefore, acceptance or rejection of the manuscript will depend on the completeness of your responses included in the next, final version of the manuscript. For this reason, and to save you from any frustrations in the end I would strongly advise against returning an incomplete revision and would also understand your decision if you choose to rather seek rapid publication elsewhere at this stage.

I look forward to seeing a revised form of your manuscript as soon as possible.

Should you find that the requested revisions are not feasible within the constraints outlined here and choose, therefore, to submit your paper elsewhere, we would welcome a message to this effect.

***** Reviewer's comments *****

Referee #1 (Comments on Novelty/Model System):

Please see my comments to the authors. In general, I think this is an interesting manuscript worthy of publication in EMBO. The authors did a significant amount of work to show the the inverse relationship between CUEDC2 and GPX1, and how CUEDC2 levels impact ROS generation and cardiac injury in the context of oxidative stress, ischemia/reperfusion, and hypoxia/reoxygenation. This aspect of the manuscript appears very solid. The data demonstrating ubiquitylation of GPX1 as a function of CUEDC2 is also generally convincing, and the data purporting that CUEDC2 acts as an adaptor protein facilitating TRIM33 ubiquitylation of GPX1 is fairly solid, although I do see some room for clarification, as delineated in my comments to the authors.

There are some issues with the cardiac ischemia/reperfusion studies in vivo that should be addressed, and the gene delivery experiments (AAV9 data) also need some significant clarifications.

In consideration of the above, I still am very enthusiastic about the manuscript, and support publication after appropriate revision.

Referee #1 (Remarks):

This is a very interesting manuscript that evaluates the role of CUEDC2 in cardiac myocyte survival in several contexts (hypoxia/reoxygenation, H2O2-induced cell death, ischemia-reperfusion in vivo, and cardiac aging). The primary findings are that CUEDC2, in conjunction with TRIM33, mediates the ubiquitylation and proteosomal destruction of GPX1, and that this promotes oxidation-associated cardiomyocyte cell death in the context of the above-delineated insults. This could represent a major, previously unrecognized, pathway regulating cardiac survival, and thus could present an important new therapeutic target. There are multiple findings delineated that are of significant interest and potential importance. That said, there are specific issues that should be addressed, as below:

1. In figure 1, the decrease in CUEDC2 in response to ischemia/reperfusion and hypoxia/reoxygenation, and that this can be inhibited by proteasome inhibition, is interesting. Whereas dependency on TRIM33 for GPX1 ubiquitylation and proteasome destruction is addressed later in the manuscript, the potential mechanism and responsible ubiquitin ligase responsible for CUEDC2 destruction is not addressed, but would seem to be of high significance - especially if a potential therapeutic goal is to reduce CUEDC2 levels in the context of myocardial injury. Perhaps this has been addressed in previous work? If so, this should be made clear in the manuscript, and if not, this should at least be discussed.

2. It is unclear from the human heart data (figure 1) whether what is being shown is actually the 'border zone' or the infarcted tissue. Also, what is the time from the infarction to death in the patients who suffered the MI? This is not included in Supplemental Table 1, and would be important in assessing the relevance of the data with regards the possibility and timing of intervention. I

understand that it may be difficult to obtain some of this data, given potential restrictions on patient information, but it would be good to include this.

3. In the above context (i.e. comment 2), did you look at GPX1 expression in the border zone? In the mouse experiments did the reduction in CUEDC2 after I/R occur globally, or only regionally in the area at risk? Was this reduction accompanied by a local increase in GPX1?

4. The data demonstrating the inverse relationship between GPX1 and CUEDC2, the half-life data demonstrating a marked prolongation of half-life of GPX1 in the absence of CUEDC2, the data showing that inhibition of proteasomal degradation abrogated the inverse relationship between CUEDC2 and GPX1, and the ubiquitylation data are well-done and convincing. The mutation of the ring portion of TRIM33 is a nice approach, but the data shown are not entirely convincing (figure 6C). It would be helpful to show that shRNA knockdown of TRIM33 also abrogated the inverse relationship between CUEDC2 and GPX1.

5. The data does support the presumption that CUEDC2 is a required adaptor for TRIM33-mediated ubiquitylation of GPX1, but it remains unclear how CUEDC2 facilitates TRIM33 ubiquitylation of GPX1. Although this reviewer understands this is a question for future studies, this purported adaptor function, and potential mechanistic underpinnings should be at least briefly addressed in the discussion - given that this is a major mechanistic claim of the manuscript. Also, it would I think be helpful to include a schematic delineating the relationships between CUEDC2, TRIM33, and GPX1 concluded from the findings.

6. Although the data presented in the graphs is convincing, the pictures of the heart sections in figure 7 are confusing. The sham-treated (no LAD ligation per the methods section) hearts for instance appear to show differential staining with the Alcian blue (e.g. significant staining of the RV free wall in the WT heart on the left). Perhaps the methods were misstated, and the ligature was briefly placed on the LAD during Alcian blue injection, without the 30 minute ischemia period. If so, this should be clarified.

7. Assessment of LV chamber size and contractile function after only 1 week post-ischemia reperfusion is a bit early for accurate analysis. Also, the increase in LV internal diameter in diastole (nearly double) is fairly dramatic for an infarction that is less than 30% of the LV, and again, the cardiac sections shown do not seem congruent with this echocardiographic data.

7. Although the authors state that AAV9 is cardiotropic, they should show the extent of gene transfer at 4 weeks (e.g. GFP fluorescence for the controls, or immunostaining for GFP if autofluorescence of the myocardium is problematic, as it can be).

8. The immunoblot in 8A is of poor quality compared to all the others shown in the manuscript. This might be a function of the PDF I received.

Referee #2 (Comments on Novelty/Model System):

The study by Jian et al. is very well conducted, using a variety of molecular biological, physiological and histological methods to decipher the mechanisms underlying the interplay between Cuedc2 and GPX1 protein expression and the ensuing consequences for tissue damage during ischemia/reperfusion and cardiac function. The quality of the experiments looks sound, adequate methods were applied and the conclusions from the results are logical.

To my knowledge, this is the first study to report a role of CUEDC2 in the regulation of redox-dependent tissue damage. Based on the identified mechanism, CUEDC2 may in fact be a novel therapeutic target for ischemic heart disease and/or aging, although a defined strategy to apply this in a model or even in humans is still missing (for instance, by using a specific inhibitor or a gene targeting therapy etc.). Thus, in my opinion, the manuscript has the novelty and overall scientific priority to be published in EMBO Molecular Medicine and therefore matches the scope of the

journal.

Referee #2 (Remarks):

The present study by Jian et al. investigates the role of CUEDC2 in the heart. CUEDC2 is a protein involved in ubiquitination of several targets and is thereby a regulator in inflammation and cancer. In the present study, the authors - who have already done long-standing research on this protein in the above mentioned areas - now find that CUEDC2 also controls the ubiquitination of glutathione peroxidase (GPX) in the heart, which is required for the anti-oxidative defense. They observe that after ischemia/reperfusion, CUEDC2 is downregulated, which increases the expression of GPX (due to less ubiquitination). When deleting CUEDC2, GPX expression is upregulated already under baseline conditions, and in these mice, myocardial I/R injury and remodeling as well as aging are substantially ameliorated. The study uses a great variety of methods *in vitro* and *in vivo* to mechanistically proof the causality of this novel mechanism.

There are some issues that the authors need to address:

- 1.) It is not clear how the human tissue was obtained. If these are samples from patients with acute myocardial infarction, were they obtained after the patient died? This must have been post-mortem, which is problematic since also non-infarcted area would be affected post-mortem. Usually, human heart samples are obtained during heart transplantations, but these hearts would not be acutely ischemic but from patients with ischemic cardiomyopathy. There should also be more information on other patient parameters if these samples come from a commercial company. And checking their homepage, I do not find any human myocardium offered. Please specify the source of these samples more clearly and how the tissue was obtained.
- 2.) Figure 1: You have to be more precise about the differentiation of ischemia and ischemia/reperfusion. During ischemia alone, due to the lack of oxygen, ROS production is not typically elevated (this is still a bit controversial, but certainly any ROS production would be expected to be low). ROS production occurs especially during reperfusion. In Figure 1B, you show CUEDC2 expression at time points 0, 3, 6 and 12 hours. What you do NOT show is how ischemia per se affected the expression, since for that you need to compare time point "0" (which according to your scheme is after 30 min of ischemia) with the expression before ischemia. Please take care that also in the legend in the text, you are clear whether you are looking at ischemia (which you are not) or ischemia/reperfusion.
- 3.) Is this a global or a cardiomyocyte-specific KO of CUEDC2? If global, you should acknowledge that at least *in vivo*, the mechanisms may be mediated by cell types other than cardiac myocytes as well.
- 4.) Fig. S1A: Do the animals develop cardiac hypertrophy or show increased ANP or BNP expression after 2 weeks of TAC? Why have you chosen 2 weeks of TAC? How do you explain the different results regarding MI vs. TAC?
- 5.) It may be helpful to draw a scheme of the mechanisms in effect for making the findings more comprehensible, in particular the interaction of CUEDC2 with the proteasome response.
- 6.) Were the experiments in Figure 4A performed in isolated cardiomyocytes or lysates from whole hearts (left or right ventricle? Both ventricles?). The description is somewhat confusing (heart in the manuscript, but cardiomyocytes in the Figure legend).
- 7.) In your manuscript you refer to Fig S8, but it was not available. Please add!
- 8.) You show that deleting CUEDC2 is protective in I/R injury or aging, while under normal conditions, mice are normal. Philosophically, why is it existing then at all? What other pathways are controlled through CUEDC2? If developing a therapeutic strategy to inhibit CUEDC2, what other cellular processes could be expected to be affected, and what consequence could be expected in other organs?
- 9.) Why is it more useful to delete CUEDC2 instead of overexpressing GPX?

Referee #3 (Remarks):

The paper entitled "CUEDC2 modulates cardiomyocytes oxidative capacity by regulating GPX1 stability" submitted by Jian et al. seeks to define the underlying molecular mechanism through

which loss of CUEDC2 improves survival of the myocardium following I/R. The *in vivo* data showing a reduction in infarct size following I/R in *Cuedc2*^{-/-} animals makes a good case that this protein is relevant to cardiac biology and therefore of value to study. Unfortunately the molecular studies that claim to identify TRIM33-mediated degradation of GPX1 as the relevant mechanism of action are incomplete, poorly described and often not properly controlled.

Major issues:

1. Through out the manuscript many of the experiments are insufficiently described and appear to be inconsistent with one another. This is particularly the case with the tissue culture "I/R" experiments. A wide range of protocols appears to have been used and never clearly defined. The figure 2 legend refers to 4 hours "anoxia" (0.3%) followed by reoxygenation (21%). In the methods "hypoxia" treatment is described as culturing in serum-free media under "hypoxic" conditions (0.3%). In some experiments rat NRVMs are used, at other times mouse NMVMs, but in neither case do the authors demonstrate that their treatment is causing cell death. Placing neonatal myocytes under hypoxic conditions in general does not damage them. Standard simulated I/R protocols require removing glucose and providing a slightly acidic conditions to more closely mimic the infarcted heart. It is doubtful that the conditions defined in the paper are sufficient to kill NRVMs. For some experiments the authors resort to adding an undefined amount of H₂O₂ to model I/R, but although H₂O₂ can indeed kill cells, it is not a valid model of I/R. Also lacking in many experiments are time courses for control oxygenated cells to verify that the changes being tracked are set in motion by the I/R treatment, and not simply changes that occur in conjunction with maturation of isolated neonatal cells (ie. in Fig1C, although this occurs through out the paper).
2. In the analysis of heart tissue in the animal experiments the region of the heart being analyzed is not defined. Are samples from the infarct, border, or remote zone?
3. It is important that changes in GPX1 and CUEDC2 mRNA levels be assessed in these experiments, particularly since the CUEDC2 has been shown capable of influencing transcription.
4. The implication of this paper is that CUEDC2/TRIM33/GPX1 undergo some kind of functional interaction that facilitates GPX1 degradation. However, there is no evidence provided of a direct interaction. This is a fundamental weakness. The text alludes to a Co-IP of CUEDC2 and GPX1 in supplemental fig 8. But I do not find a supplemental fig 8.
5. Is the effect of CUEDC2 dependent on the CUE-domain?
6. Throughout the word "expression" is used when referring to changes in protein levels. Although frequently see misused in the literature, "expression" refers to gene activity. The use of the word expression is simply confusing and implies gene activity.
7. Given that phosphorylation of CUEDC2 by ERK has been implicated in promoting CUEDC2 degradation, it is surprising that the authors have tracked JNK and p38, but ignored the ERK leg of MAPK signaling.
8. The experiments regarding the impact on changes in NFκB activity are grossly inadequate given the importance of this pathway in the immune response and CUEDC2 biology, and that the animals are a whole-body KO. If the extent of damaged tissue is less, this will indeed reduce the extent of the immune response and the number of inflammatory cells infiltrating into the damaged region. Therefore, it is impossible to separate these changes from those that loss of CUEDC2 might be mediating on the animals immune system. The luciferase assays in Fig 3C do not mention the use of a co-transfected reporter to control for differences in transfection efficiency, transcriptional squelching, or cell proliferation, therefore, they are not a reliable measure of NFκB activity. The supplementary EMSA appears to be an n=1 and provides no information regarding the area of the heart used to make the extract.
9. Fig 5A is the only experiment actually showing a change in the rate of GPX1 degradation. The experiment appears to have been done on film. The thresholding of film dictates that the intensity of the initial signal at T=0 is the same in exposure to be quantified.
10. Ubiquitin abundance is seldom rate limiting, therefore in Fig 5C GPX1 ubiquitination in lane 2 should not require transfection with the Myc-Ub to accumulate ubiquitinated species. The figure indicates it was probed with GPX1.
11. A gel of the IPed complex used for mass spec should be provided along with it's negative control. Please include at least a summary of the mass-spec results in the supplemental materials. Was CUEDC2 among the protein identified?
12. The discussion says that the work demonstrates that TRIM33-dependen ubiquitination of GPX1 is "facilitated" by CUEDC2 binding. What is the evidence for this?
13. If there is data showing that PR and ER-alpha levels are not changed in the KO, this needs to be

shown.

14. In figure 6C where is the HA-CUEDC2 coming from?

15. In Fig 7 please quantify TUNEL signal co-localized with cardiac myocytes separately from those co-localized with other cell types.

16. Figure * needs to include a similar blot showing tissue-distribution of the GFP virus.

17. In the aged mice, is there a genotype difference in the body weight of the animals?

18. In SFig1 what is the point of pretreating with MG132 for 4 hours. Please clarify the conditions used, is this anoxia or simulated I/R?

19. Something is very wrong with the data in Online Table2. The functional measures shown are consistent with sedated mice, however, the heart rates are not those of a sedated mouse. What were the heart rates during echocardiography?

1st Revision - authors' response

11 March 2016

Reviewer #1

We appreciate the reviewer's encouraging comments and constructive suggestions about our manuscript. The reviewer indicated that the effect of CUEDC2 in the heart could represent a major, previously unrecognized, pathway regulating cardiac survival, and thus could present an important new therapeutic target. The reviewer also raised several concerns and gave us some suggestions to further improve our manuscript. Accordingly, we performed additional experiments and we have further confirmed that CUEDC2 promoted TRIM33 mediated GPX1 ubiquitination and proteasome-dependent degradation, modulating the antioxidant capacity of cardiomyocytes. Also, we have provided an in-depth discussion of the unique role of CUEDC2 in antioxidant defence.

Below are our point-to-point responses to the reviewer's concerns.

1) In figure 1, the decrease in CUEDC2 in response to ischemia/reperfusion and hypoxia/reoxygenation, and that this can be inhibited by proteasome inhibition, is interesting. Whereas dependency on TRIM33 for GPX1 ubiquitination and proteasome destruction is addressed later in the manuscript, the potential mechanism and responsible ubiquitin ligase responsible for CUEDC2 destruction is not addressed, but would seem to be of high significance - especially if a potential therapeutic goal is to reduce CUEDC2 levels in the context of myocardial injury. Perhaps this has been addressed in previous work? If so, this should be made clear in the manuscript, and if not, this should at least be discussed.

Response: The reviewer raised an important point. Previously, we reported that CUEDC2 protein could be degraded under UV-treatment, and phosphorylation of CUEDC2 facilitates this degradation (Zhang *et al. Proc Natl Acad Sci USA*. 2013). In this study, the decrease in CUEDC2 in response to ischemia/reperfusion is not at mRNA level. We postulated that the post-translational modification might regulate the stability of CUEDC2. We have identified some E3 ligase candidates for CUEDC2 degradation by immunoprecipitation & mass spectrometry analysis. The potential mechanism of the E3 ligase responsible for CUEDC2 destruction following I/R injury is being investigated now. This is of high significance for developing a potential therapeutic strategy to reduce CUEDC2 levels in the context of myocardial injury. We added this in the section of Discussion.

2) It is unclear from the human heart data (figure 1) whether what is being shown is actually the 'border zone' or the infarcted tissue. Also, what is the time from the infarction to death in the patients who suffered the MI? This is not included in Supplemental Table 1, and would be important in assessing the relevance of the data with regards the possibility and timing of intervention. I understand that it may be difficult to obtain some of this data, given potential restrictions on patient information, but it would be good to include this.

Response: We examined CUEDC2 levels in heart samples from myocardial infarction patients. The human heart data was from 'border zone' according to the morphological change of myocardium. As cardiomyocytes were irreversibly damaged in infarcted tissue, it was more relevant to study the changes in the border zone and to salvage the cardiomyocytes in the border zone. We added this in the section of Methods in this revised manuscript.

We fully agree with the reviewer that the time from the infarction to death in the patients who suffered from the MI would be important in assessing the relevance of the data with regards the possibility and timing of intervention. We contacted with the Biocat company where we bought the tissue array from for this information. However, this information is not available due to the strict regulations to preserve donors' anonymity in Germany. We appreciate the reviewer's understanding on the difficulty of obtaining this data.

3) *In the above context (i.e. comment 2), did you look at GPX1 expression in the border zone? In the mouse experiments did the reduction in CUEDC2 after I/R occur globally, or only regionally in the area at risk? Was this reduction accompanied by a local increase in GPX1?*

Response: The reviewer raised an important point. We examined the GPX1 levels in the border zone of the same tissue samples and found that GPX1 expression was increased when CUEDC2 was low. We added the representative figure and statistic results in Figure 4C. In the mouse model, the reduction of CUEDC2, accompanied by an increase in GPX1, was regionally in the area at risk (Fig. 4B).

4) *The data demonstrating the inverse relationship between GPX1 and CUEDC2, the half-life data demonstrating a marked prolongation of half-life of GPX1 in the absence of CUEDC2, the data showing that inhibition of proteosomal degradation abrogated the inverse relationship between CUEDC2 and GPX1, and the ubiquitination data are well-done and convincing. The mutation of the ring portion of TRIM33 is a nice approach, but the data shown are not entirely convincing (figure 6C). It would be helpful to show that shRNA knockdown of TRIM33 also abrogated the inverse relationship between CUEDC2 and GPX1.*

Response: Thanks a lot for this constructive suggestion. We knocked down the expression of TRIM33 in primary cardiomyocytes and the inverse relationship between CUEDC2 and GPX1 was abrogated. We added this result in revised Appendix Fig. S11.

5) *The data does support the presumption that CUEDC2 is a required adaptor for TRIM33-mediated ubiquitination of GPX1, but it remains unclear how CUEDC2 facilitates TRIM33 ubiquitination of GPX1. Although this reviewer understands this is a question for future studies, this purported adaptor function, and potential mechanistic underpinnings should be at least briefly addressed in the discussion - given that this is a major mechanistic claim of the manuscript. Also, it would I think be helpful to include a schematic delineating the relationships between CUEDC2, TRIM33, and GPX1 concluded from the findings.*

Response: Thanks a lot for the reviewer's constructive suggestion and understanding. Here we confirmed that CUEDC2 plays indispensable role in TRIM33-mediated ubiquitination of GPX1. CUEDC2 protein contains a CUE domain (Coupling of ubiquitin conjugation to endoplasmic reticulum degradation domain). It was reported that CUE is a kind of ubiquitin-binding domains (UBDs), which bind multiple ubiquitin molecules and promote the ubiquitination. Our new data showed that the CUE domain was indispensable for the regulation of CUEDC on GPX1 (Fig. 5E and Appendix Fig. S9B). Thus CUEDC2 might facilitate TRIM33-mediated ubiquitination of GPX1 by binding multiple ubiquitin molecules. We added this in the section of Discussion. Also, According to the reviewer's suggestion, we provided a schematic mechanism delineating the relationships between CUEDC2, TRIM33, and GPX1 and added it in the Appendix Fig. S14.

6) *Although the data presented in the graphs is convincing, the pictures of the heart sections in figure 7 are confusing. The sham-treated (no LAD ligation per the methods section) hearts for instance appear to show differential staining with the Alcian blue (e.g. significant staining of the RV free wall in the WT heart on the left). Perhaps the methods were misstated, and the ligature was briefly placed on the LAD during Alcian blue injection, without the 30 minute ischemia period. If so, this should be clarified.*

Response: We thank the reviewer for the careful reading. As the reviewer's description, in sham group, the ligature was briefly placed on the LAD during Alcian blue injection in order to define the area at risk, without the 30 minute ischemia period. We clarified this information in the detailed methods online.

7) *Assessment of LV chamber size and contractile function after only 1 week post-ischemia reperfusion is a bit early for accurate analysis. Also, the increase in LV internal diameter in diastole (nearly double) is fairly dramatic for an infarction that is less than 30% of the LV, and again, the cardiac sections shown do not seem congruent with this echocardiographic data.*

Response: The reviewer raised an important point. Actually, we examined LV chamber size and contractile function at 1 week, 2 weeks and 4 weeks post-I/R. The results are shown below for the reviewer. We found that there was significant difference between WT group and *Cuedc2*^{-/-} at 1 week post-ischemia reperfusion and this difference was persistent at 2 weeks and 4 weeks post-I/R. So, we showed the data at the time point of 1 week.

We thank the reviewer for pointing out this issue. We realized that we made a mistake in this figure. The data shown in the Fig.7E, right panel, about LV internal diameter was actually in the phase of systole, not in the phase of diastole. We've corrected this mistake in the revised manuscript. There's a significant increase in LVID.s following I/R injury due to the myocardial cell death and reduced contractile function. *Cuedc2* ablation led to a significant decrease in LVID.s compared with their WT littermates. The diastolic LVID, however, didn't change after 1 week post-I/R (shown below).

The reviewer pointed out that cardiac sections shown do not seem congruent with this echocardiographic data. The cardiac sections were obtained at 24 h post-I/R, and the echocardiographic data was obtained at 1 week post-I/R. The data presented in these experiments from different time points may cause the confusion. We included the time point information in the figure legends.

8) *Although the authors state that AAV9 is cardiotropic, they should show the extent of gene transfer at 4 weeks (e.g. GFP fluorescence for the controls, or immunostaining for GFP if autofluorescence of the myocardium is problematic, as it can be).*

Response: We examined the GFP expression in control group by Western blot assay to show the extent of gene transfer at 4 weeks. The expression of GFP was particularly high in heart which is similar with that of CUEDC2. We showed this data in Appendix Fig. S12.

9) *The immunoblot in 8A is of poor quality compared to all the others shown in the manuscript. This might be a function of the PDF I received.*

Response: We replaced this immunoblot with the one of higher resolution.

Reviewer #2

The reviewer indicated “this is the first study to report a role of CUEDC2 in the regulation of redox-dependent tissue damage. Based on the identified mechanism, CUEDC2 may in fact be a novel therapeutic target for ischemic heart disease and/or aging”. Meanwhile, the reviewer raised some concerns and suggestions for further improvement of our manuscript. We appreciate the reviewer’s encouraging comments on our manuscript. We have performed additional experiments as suggested and have re-written the manuscript to further clarify our results.

Below are our point-by-point responses to the reviewer’s concerns.

1) *It is not clear how the human tissue was obtained. If these are samples from patients with acute myocardial infarction, were they obtained after the patient died? This must have been post-mortem, which is problematic since also non-infarcted area would be affected post-mortem. Usually, human heart samples are obtained during heart transplantations, but these hearts would not be acutely ischemic but from patients with ischemic cardiomyopathy. There should also be more information on other patient parameters if these samples come from a commercial company. And checking their*

homepage, I do not find any human myocardium offered. Please specify the source of these samples more clearly and how the tissue was obtained.

Response: We obtained the human heart tissue array from BioCat GmbH, Heidelberg, Germany (Cat No. MYO1301-5-OL). These samples were also used in other study (Fan et al. *Nat Cell Biol.* 2012, Fig 5a). The sample information on the age, gender, diagnosis, localization, etc. is available on the company website (<http://www.biocat.com/products/MYO1301-5-OL>). We contacted with the company for more patients' information and unfortunately it's not available due to the strict regulations to preserve donors' anonymity in Germany. We have no information about how timely those acute myocardial infarction samples were collected after the patients died. Nevertheless, we found that the protein level of CUEDC2 was significantly reduced only in the ischemic border zone in the acute myocardial infarction samples, but not in the distant zone (Appendix Fig. S3), suggesting that the change of CUEDC2 level was mainly due to the acute infarction.

2) Figure 1: You have to be more precise about the differentiation of ischemia and ischemia/reperfusion. During ischemia alone, due to the lack of oxygen, ROS production is not typically elevated (this is still a bit controversial, but certainly any ROS production would be expected to be low). ROS production occurs especially during reperfusion. In Figure 1B, you show CUEDC2 expression at time points 0, 3, 6 and 12 hours. What you do NOT show is how ischemia per se affected the expression, since for that you need to compare time point "0" (which according to your scheme is after 30 min of ischemia) with the expression before ischemia. Please take care that also in the legend in the text, you are clear whether you are looking at ischemia (which you are not) or ischemia/reperfusion.

Response: This is quite a good point. According to the reviewer's suggestion, we examined the protein levels of CUEDC2 in the sham group and I/R treatment. As the time point "0" represent the 30-minute after ischemia, the sham group is the heart without ischemia. After we compared these two groups, we found that there was no significant difference in CUEDC2 protein level, indicating that ischemia 30-minute alone had no effect on CUEDC2 expression level (in revised Appendix Fig. S2A). We thank the reviewer for the kind reminder on the clear description of ischemia/reperfusion. We made some changes in the revised manuscript accordingly.

3) Is this a global or a cardiomyocyte-specific KO of CUEDC2? If global, you should acknowledge that at least in vivo, the mechanisms may be mediated by cell types other than cardiac myocytes as well.

Response: *Cuedc2* gene was globally knocked out in the mouse. We showed that the cardiomyocytes isolated from *Cuedc2*^{-/-} mice were protective from oxidative stress stimulation and hypoxia/reoxygenation (H/R) injury. In addition, in the heart tissue during I/R injury as well as cardiomyocytes treated with H/R, we showed a reduction of CUEDC2 level. In cardiomyocytes, knockout of CUEDC2 increases GPX1 level, decreases the ROS level and consequent cell death. These results suggest the CUEDC2 loss in the cardiomyocytes is an important intrinsic mechanism of protecting heart from I/R injury. Nevertheless, as the reviewer pointed out, we could not fully rule out the possibility that the loss of CUEDC2 in other cell types might contribute to the heart protection. We've added this discussion in the revised manuscript.

4) Fig. S1A: Do the animals develop cardiac hypertrophy or show increased ANP or BNP expression after 2 weeks of TAC? Why have you chosen 2 weeks of TAC? How do you explain the different results regarding MI vs. TAC?

Response: After we tested the cardiac hypertrophy by echocardiography at different time (1 week, 2 weeks, 4 weeks) post-TAC. As previously reported, 2 weeks after TAC, the mouse heart developed significantly cardiac hypertrophy and the *Myh7*, *Anp*, *Bnp*, *Acta1* levels were all increased when compared to the sham group. We added this data in the revised supplemental Fig. 1A and 1B.

We tested the CUEDC2 protein levels in the heart upon acute I/R injury, which leads to MI, or chronic overload stress by TAC surgery. I/R is an acute hypoxic stress followed by reperfusion, whereas TAC is a chronic hypoxic stress. The decrease of CUEDC2 protein level only occurred during reperfusion process. Ischemia alone didn't result in the degradation of CUEDC2 protein in heart (in revised Appendix Fig. S2A). In addition, we are sorry for the mislabeling in Appendix Fig. S2D and for the confusion that might cause. "Anoxia" should be replaced by "hypoxia/reoxygenation", a condition that simulated I/R in vitro.

5) *It may be helpful to draw a scheme of the mechanisms in effect for making the findings more comprehensible, in particular the interaction of CUEDC2 with the proteasome response.*

Response: We thank the reviewer for the constructive advice. We drew a scheme of the mechanisms and added it in supplementary Fig 14.

6) *Were the experiments in Figure 4A performed in isolated cardiomyocytes or lysates from whole hearts (left or right ventricle? Both ventricles?). The description is somewhat confusing (heart in the manuscript, but cardiomyocytes in the Figure legend).*

Response: In Figure 4A, we extracted the protein sample from the left ventricle. We are sorry for the unclear description. We have amended this in the detailed methods online and Figure legend accordingly.

7) *In your manuscript you refer to Fig S8, but it was not available. Please add!*

Response: We are sorry for the mistake. “Fig S8” should be Fig S7. We have modified in this revised manuscript.

8) *You show that deleting CUEDC2 is protective in I/R injury or aging, while under normal conditions, mice are normal. Philosophically, why is it existing then at all? What other pathways are controlled through CUEDC2? If developing a therapeutic strategy to inhibit CUEDC2, what other cellular processes could be expected to be affected, and what consequence could be expected in other organs?*

Response: CUEDC2 is a multi-functional protein and is involved in the regulation of many key cellular and physiological events. In our previous studies, we found that CUEDC2 functions as a negative regulator of NF- κ B signaling to prevent inflammatory damage (Li et al, *Nat Immunol.* 2008). CUEDC2 KO mice are more prone to colitis-associated cancer (Chen et al, *Cell Rep.* 2014). CUEDC2 dysregulation might contribute to tumor development by causing chromosomal instability (Gao, et al, *Nat Cell Biol.* 2011), suggesting a physiological role of CUEDC2 in protecting against tumorigenesis. CUEDC2 also promotes the degradation of PR and ER α by affecting the ubiquitin-proteasome pathway and confers endocrine resistance in breast cancer (Zhang, et al, *EMBO J.* 2007; Pan, et al, *Nat Med.* 2011). In addition, high CUEDC2 protein level in heart might be important to maintain the homeostasis of GPX1 protein under normal conditions. Given that CUEDC2 plays various roles under different conditions, the heart-specific delivery should be taken into account to avoid the potential side effects in other organs if developing a therapeutic strategy to inhibit CUEDC2 for I/R injury protection. We added this discussion in the revised manuscript.

9) *Why is it more useful to delete CUEDC2 instead of overexpressing GPX?*

Response: The reviewer raised an important point. ROS plays critical roles in I/R injury. Therefore, scavenging ROS by overexpressing GPX1 could protect the heart from I/R injury. However, GPX1 is ubiquitously expressed in all tissues and the overexpression of GPX1 could cause aberrant ERK activation (Autheman et al, *Pediatr Res.* 2012), insulin resistance and obesity (McClung et al, *Proc Natl Acad Sci USA.* 2012). Therefore, direct overexpressing GPX1 might not be an ideal therapeutic strategy for heart diseases. To down regulate the cardiac specifically highly expressed protein, such as CUEDC2, might be a potential effective strategy to protect I/R injury, without interfering with the physiological function. We added this information in the introduction in the revised manuscript.

Reviewer #3

We appreciate the reviewer’s encouraging comments and constructive suggestions about our manuscript. The reviewer commented that CUEDC2 was relevant to cardiac biology and therefore of value to study. The reviewer also raised some concerns and suggestions to further improve our manuscript. According to the reviewer’s suggestions, we performed additional experiments and revised the manuscript.

Below are our point-to-point responses to the reviewer’s concerns.

1) Throughout the manuscript many of the experiments are insufficiently described and appear to be inconsistent with one another. This is particularly the case with the tissue culture "I/R" experiments. A wide range of protocols appears to have been used and never clearly defined. The figure 2 legend refers to 4 hours "anoxia" (0.3%) followed by reoxygenation (21%). In the methods "hypoxia" treatment is described as culturing in serum-free media under "hypoxic" conditions (0.3%). In some experiments rat NRVMs are used, at other times mouse NMVMs, but in neither case do the authors demonstrate that their treatment is causing cell death. Placing neonatal myocyte under hypoxic conditions in general does not damage them. Standard simulated I/R protocols require removing glucose and providing a slightly acidic conditions to more closely mimic the infarcted heart. It is doubtful that the conditions defined in the paper are sufficient to kill NRVMs. For some experiments the author s resort to adding an undefined amount of H₂O₂ to model I/R, but although H₂O₂ can indeed kill cells, it is not a valid model of I/R. Also lacking in many experiments are time courses for control oxygenated cells to verify that the changes being tracked are set in motion by the I/R treatment, and not simply changes that occur in conjunction with maturation of isolated neonatal cells (ie. in Fig1C, although this occurs throughout the paper).

Response: The reviewer raised an important point. All the hypoxic conditions in our study is 0.3% oxygen treatment. We amended the incorrect description of 'anoxia' in the revised manuscript. As the reviewer pointed out, removing glucose and providing a slightly acidic conditions is a standard method to mimic the infarcted heart *in vitro* (Xie et al. *Circulation*. 2014). In addition, hypoxia accompanying with serum-free media, followed by reoxygenation accompanying with adding back serum is an alternative method to mimic I/R injury *in vitro* (Shibata et al. *Nat Med*. 2005; Oshima et al. *Circulation*. 2008; Zhang et al. *Nat Med*. 2016). Following this protocol, we examined the cell death of cardiomyocytes under the H/R treatment, and found that cell death significantly increased in both WT and *Cuedc2*^{-/-} cardiomyocytes (Appendix Fig. S5C).

Under I/R injury, oxidative stress mediated by a large amount of reactive oxygen species is the main cause of heart dysfunction. To examine whether CUEDC2 ablation protects the cells from oxidative stress damage, we treated cardiomyocytes with H₂O₂ and found a significant higher percentage of *Cuedc2*^{-/-} cardiomyocytes survived the treatment. This result suggests that CUEDC2 might play important roles in the process of reactive oxygen species generation or elimination.

We thank the reviewer for the constructive advice. To verify the changes of CUEDC2 levels being tracked are set in motion by I/R treatment, we performed an additional experiment with control oxygenated cells at the similar time courses. The result showed that CUEDC2 protein levels do not alter in conjunction with maturation of isolated neonatal cells. We added this data in the revised Appendix Fig. S2C.

2) In the analysis of heart tissue in the animal experiments the region of the heart being analyzed is not defined. Are samples from the infarct, boarder, or remote zone?

Response: In the analysis of heart tissue in the animal experiments, the samples were from the area at risk following I/R treatment. This area includes the infarct and border zone. After I/R treatment, the left ventricle was divided into the area at risk and normal area. Myocardial infarction only occurred in the area at risk. Therefore, we analyzed the area at risk for the differences between WT and *Cuedc2*^{-/-} mice to assess the role of CUEDC2 in protecting heart from I/R injury. We included this important information in the Method section in the revised manuscript.

3) It is important that changes in GPX1 and CUEDC2 mRNA levels be assessed in these experiments, particularly since the CUEDC2 has been shown capable of influencing transcription.

Response: According to the reviewer's suggestion, we examined the mRNA levels of CUEDC2 and GPX1 during I/R injury. We found that neither of the mRNA levels of CUEDC2 and GPX1 changed during I/R injury. These results suggest that CUEDC2 does not influence transcription of GPX1. We have added these data in the Appendix Fig. S2B and Fig. S8B.

4) The implication of this paper is that CUEDC2/TRIM33/GPX1 undergo some kind of functional interaction that facilitates GPX1 degradation. However, there is no evidence provided of a direct interaction. This is a fundamental weakness. The text alludes to a Co-IP of CUEDC2 and GPX1 in supplemental fig 8. But I do not find a supplemental fig 8.

Response: We are sorry for the mistake. "Supplemental Fig. 8" should be "Supplemental Fig. 7" in the original manuscript. This result was moved to Fig. 5E in the revised manuscript, which showed that CUEDC2 interacts with GPX1. Our new data showed that CUEDC2 did not affect the

interaction of GPX1 and its E3 ligase TRIM33. We added this result in the revised Appendix Fig. S11B.

5) *Is the effect of CUEDC2 dependent on the CUE-domain?*

Response: The reviewer raised an important point. We constructed a CUE domain deletion mutant of CUEDC2 (CUEDC2 Δ ^{CUE}), and tested the effect of this mutant CUEDC2 on GPX1. We found that the CUE domain was indispensable for the regulation of CUEDC on GPX1, as CUEDC2 Δ ^{CUE} did not interact with GPX1 and GPX1 was not decreased when CUEDC2 Δ ^{CUE} was overexpressed. We added this data in the Fig. 5E and Appendix Fig. S9B.

6) *Throughout the word "expression" is used when referring to changes in protein levels. Although frequently see misused in the literature, "expression" refers to gene activity. The use of the word expression is simply confusing and implies gene activity.*

Response: Thanks a lot for the reviewer's comments. We changed "expression" to "protein level" in revised manuscript.

7) *Given that phosphorylation of CUEDC2 by ERK has been implicated in promoting CUEDC2 degradation, it is surprising that the authors have tracked JNK and p38, but ignored the ERK leg of MAPK signaling.*

Response: In our study, we subjected the heart to 30-min of ischemia and followed by 30-min reperfusion, and detected the phosphorylation of JNK, p38 as well as ERK MAPK in WT and *Cuedc2*^{-/-} heart. At this time point, we found that JNK and p38 were significantly activated whereas ERK1/2 was hardly activated in response to I/R injury. Therefore, we did not show ERK1/2 result in our previously manuscript. Now we included this result in Appendix Fig. S6 in the revised manuscript.

8) *The experiments regarding the impact on changes in NFkB activity are grossly inadequate given the importance of this pathway in the immune response and CUEDC2 biology, and that the animals are a whole-body KO. If the extent of damaged tissue is less, this will indeed reduce the extent of the immune response and the number of inflammatory cells infiltrating into the damaged region. Therefore, it is impossible to separate these changes from those that loss of CUEDC2 might be mediating on the animals immune system. The luciferase assays in Fig 3C do not mention the use of a co-transfected reporter to control for differences in transfection efficiency, transcriptional squelching, or cell proliferation, therefore, they are not a reliable measure of NF-kB activity. The supplementary EMSA appears to be an n=1 and provides no information regarding the area of the heart used to make the extract.*

Response: We previously reported that CUEDC2 was a negative regulator of TNFa/ IL-1b induced NF- κ B activation (Li et al, *Nat Immunol.* 2008). Consistently, in Fig. 3C, when CUEDC2 was knocked out, TNFa induced NF- κ B activation was upregulated. Therefore, loss of CUEDC2 results in increased immune response. However, under I/R injury, NF- κ B is mainly activated by reactive oxygen species (Gordon et al, *Circ Res*, 2011; Nakano et al, *Cell Death Differ.* 2005). In our study, H/R induced NF- κ B activation was inhibited by CUEDC2 knockout. As the ablation of CUEDC2 decreases the H/R-induced ROS level (Fig.2C), the ROS downstream NF- κ B signaling pathways were subsequently inhibited (Fig. 3C).

We are sorry for unclear method description of luciferase assay. We co-transfected a control reporter Renilla for normalizing the transfection efficiency and cell proliferation, etc in all of the luciferase assays. We added this important information in the detailed method online in the revised manuscript.

For EMSA assay, we used the area at risk of LV and the result shown in Appendix Fig. S7A was a representative one of three independent experiments, and the statistic result was added in the Appendix Fig. S7B. We added this information in the revised manuscript.

9) *Fig 5A is the only experiment actually showing a change in the rate of GPX1 degradation. The experiment appears to have been done on film. The thresholding of film dictates that the intensity of the initial signal at T=0 is the same in exposure to be quantified.*

Response: As shown in the original Fig. 4A and 5A, CUEDC2 knockout significantly increases the GPX1 protein level. According to the reviewer's suggestion, we compared the rate of GPX1 degradation in WT and *Cuedc2*^{-/-} cells with different exposure films for the same intensity of initial signal at T=0 for GPX1 levels. We got a similar result with Fig. 5A, right panel. However, we

realized that the different exposures might lead to the same basal level of GPX1 protein in WT and *Cuedc2*^{-/-} cells, which is inconsistent with the result shown in Fig. 4A. Therefore, we retained the original Fig. 5A to avoid the possible confusion.

10) Ubiquitin abundance is seldom rate limiting, therefore in Fig 5C GPX1 ubiquitination in lane 2 should not require transfection with the Myc-Ub to accumulate ubiquitinated species. The figure indicates it was probed with GPX1.

Response: In each transfected cell, Flag-GPX1 protein level is much higher than the endogenous protein level. The endogenous ubiquitin is not abundant enough for sufficient exogenous GPX1 ubiquitination. We tried GPX1 ubiquitination without ubiquitin transfection. GPX1 ubiquitination could be hardly detected. Therefore, we had to transfect Myc-Ub to detect the ubiquitination.

11) A gel of the IPed complex used for mass spec should be provided along with its negative control. Please include at least a summary of the mass-spec results in the supplemental materials. Was CUEDC2 among the protein identified?

Response: According the reviewer's suggestion, we included the original gel result of the IPed complex used for mass spec in the revised Appendix Fig. S10. We also included a summary of the mass-spectrometry results in the supplementary table 4. Indeed, CUEDC2 is one of the identified proteins that bind to GPX1.

12) The discussion says that the work demonstrates that TRIM33-dependent ubiquitination of GPX1 is "facilitated" by CUEDC2 binding. What is the evidence for this?

Response: We are sorry for the inaccurate statement. We changed it in the revised manuscript as follow: We showed that the protein levels of GPX1 could be modulated by TRIM33-dependent degradation, a process that was facilitated by CUEDC2. CUEDC2 protein contains a CUE domain (Coupling of ubiquitin conjugation to endoplasmic reticulum degradation domain). It was reported that CUE is a kind of ubiquitin-binding domains (UBDs), which bind multiple ubiquitin molecules and promote the ubiquitination. As the CUE domain was indispensable for the regulation of CUEDC2 on GPX1, CUEDC2 might facilitate TRIM33-mediated GPX1 degradation by binding multiple ubiquitin molecules.

13) If there is data showing that PR and ER-alpha levels are not changed in the KO, this needs to be shown.

Response: We included this data in the Appendix Fig. S15 in the revised manuscript.

14) In figure 6C where is the HA-CUEDC2 coming from?

Response: We are sorry for the mislabeling. HA-CUEDC2 should be HA-TRIM33. We changed this in the revised manuscript.

15) In Fig 7 please quantify TUNEL signal co-localized with cardiac myocytes separately from those co-localized with other cell types.

Response: We thank the reviewer for the constructive suggestion. We marked the cardiomyocytes with specific marker Troponin I, and we add this co-localization of Troponin I and TUNEL in the revised Figure 7B.

16) Figure 8 needs to include a similar blot showing tissue-distribution of the GFP virus.

Response: In the control group, we tested GFP accordingly, and we found that the distribution of GFP was specifically in heart which is consistent with that of CUEDC2. And we added this data in the revised Appendix Fig. S12.

17) In the aged mice, is there a genotype difference in the body weight of the animals?

Response: There was no significantly difference in body weights between the aged WT or *Cuedc2*^{-/-} mice. We included this data in the revised Appendix Fig. S13.

18) In SFig1 what is the point of pretreating with MG132 for 4 hours. Please clarify the conditions used, is this anoxia or simulated I/R?

Response: In order to explore whether the reduced CUEDC2 level following I/R is due to the proteasome-dependent degradation, we pretreated the cells with MG132 for 4 hours to block ubiquitin proteasome-dependent degradation. We also performed this experiment with 1 hour MG132 pretreatment and got similar results. The MG132 pretreated cells were then subjected to

hypoxia/reoxygenation, a condition that simulated I/R in vitro, for another 2 hours. The conditions used for this experiment have been clarified in the revised manuscript.

19) Something is very wrong with the data in Online Table2. The functional measures shown are consistent with sedated mice, however, the heart rates are not those of a sedated mouse. What were the heart rates during echocardiography?

Response: Thanks a lot for this reminder. We analyzed the heart function using Vevo 770 high-resolution system, which is not equipped with an ECG module. Therefore, the heart rate could not be automatically recorded when analyzing the cardiac function. The heart rate data in original online table 2 was acquired under basic conditions, rather than under sedation conditions. The heart rates of these sedated mice could be calculated according to the original echocardiography image. The heart rates were not significantly different in the WT and *Cuedc2*^{-/-} mice (Heart rate: 424.15 ± 26.43 bpm in WT and 418.22 ± 22.26 bpm in *Cuedc2*^{-/-} mice). We included this data in the revised online table 2. To further verify this result, we measured the heart rates of sedated mice with a recently obtained new Vevo 770 high-resolution system which is equipped with ECG module and got similar results.

2nd Editorial Decision

26 April 2016

Thank you for the submission of your revised manuscript to EMBO Molecular Medicine. We have now received the enclosed reports from the referees that were asked to re-assess it. As you will see the reviewers are now globally supportive and I am pleased to inform you that we will be able to accept your manuscript pending the following final amendments:

1) Please address the minor text changes commented by referees 1 and 3. Please provide a letter INCLUDING the reviewer's reports and your detailed responses to their comments (as Word file).

Please submit your revised manuscript within two weeks. I look forward to seeing a revised form of.

***** Reviewer's comments *****

Referee #1 (Remarks):

Thank you for being responsive to previous comments, and for the additional data and clarifications.

Regarding the human heart data, I was just add (if you haven't already) a simple statement about the limitations of the data given the limited clinical data available about the patients from whom they were obtained.

Regarding the AAV gene transfer studies, I understand that logic of doing Western blots for GFP protein to evaluate gene transfer efficiency, but this does not tell you important information about the number/percentage of cells transduced, or the distribution of transduction. For future studies of this type I would consider using either a different marker gene that will allow histological assessment of gene transfer (e.g. nuclear targeted beta-gal), and/or using immunohistochemical analysis to delineate the distribution of gene transfer.

Referee #2 (Comments on Novelty/Model System):

The Authors have revised the manuscript and I find it now acceptable for publication.

Referee #2 (Remarks):

No further questions.

Referee #3 (Comments on Novelty/Model System):

This resubmission is substantially improved over the previous version, and the authors have taken care to address most of the reviewer's concerns. However, before recommending for publication anywhere, I highly recommend that the manuscript be gone through carefully for two items.

1. Grammatical errors that make it hard to follow meaning. For instance, the following sentence found in the abstract does not make sense: "Mechanistically, CUEDC2 promoted E3 ubiquitin ligases tripartite motif-containing 33 (TRIM33) mediated the antioxidant enzyme, glutathione peroxidase 1 (GPX1) ubiquitination and proteasome-dependent degradation."

2. Please go through all the figure legends to be sure that all essential information is included. Such as the identity of the sample (heart or cells?), timing, etc. There are a number of places, particularly in the supplemental material, where information is incomplete.

2nd Revision - authors' response

12 May 2016

Thank you for the prompt review of our manuscript titled "CUEDC2 modulates cardiomyocytes oxidative capacity by regulating GPX1 stability" (EMM-2015-06010) by Jian et al. We are glad to see that you and the reviews think that we addressed most concerns of the reviewers and our manuscript is suitable for publication in EMBO Molecular Medicine in principle. Following your and reviewers' suggestions, we have further revised our manuscript as detailed in the following point-to-point response.

Reviewer #1

1) Regarding the human heart data, I was just add (if you haven't already) a simple statement about the limitations of the data given the limited clinical data available about the patients from whom they were obtained.

Response: We thank the reviewer for the suggestion. We added this statement in the section of Materials and Methods.

2) Regarding the AAV gene transfer studies, I understand that logic of doing Western blots for GFP protein to evaluate gene transfer efficiency, but this does not tell you important information about the number/percentage of cells transduced, or the distribution of transduction. For future studies of this type I would consider using either a different marker gene that will allow histological assessment of gene transfer (e.g. nuclear targeted beta-gal), and/or using immunohistochemical analysis to delineate the distribution of gene transfer.

Response: We appreciate the reviewer's constructive suggestions and understanding. We will delineate the distribution of gene transfer in this way in future.

Reviewer #3

1) Grammatical errors that make it hard to follow meaning. For instance, the following sentence found in the abstract does not make sense: "Mechanistically, CUEDC2 promoted E3 ubiquitin ligases tripartite motif-containing 33 (TRIM33) mediated the antioxidant enzyme, glutathione peroxidase 1 (GPX1) ubiquitination and proteasome-dependent degradation."

Response: We thank the reviewer for the encouraging comments and suggestions. We changed the sentence as "Notably, CUEDC2 promoted E3 ubiquitin ligases tripartite motif-containing 33 (TRIM33)-mediated the antioxidant enzyme, glutathione peroxidase 1 (GPX1) ubiquitination and proteasome-dependent degradation". Moreover, this revised manuscript was thoroughly read and amended by a native-speaker, and some grammatical errors have been changed.

2) Please go through all the figure legends to be sure that all essential information is included. Such as the identity of the sample (heart or cells?), timing, etc. There are a number of places, particularly in the supplemental material, where information is incomplete.

Response: Thanks a lot for the reviewer's suggestion. We have added the essential information in the figure legends. We have clarified the source of samples, timing, etc. in the figure legends both in the main text and appendix material.

If there are any further questions, please do not hesitate to contact us.

We thank you once more for your re-consideration of our manuscript and the positive feed-back.

We look forward to your reply.

Corresponding Author Name: Professor Leo Dunkel

Manuscript Number: EMM-2016-06250-T